# Current Biomedical Applications of 3D-Printed Hydrogels

**DOI:** 10.3390/gels10010008

**Published:** 2023-12-21

**Authors:** Allan John R. Barcena, Kashish Dhal, Parimal Patel, Prashanth Ravi, Suprateek Kundu, Karthik Tappa

**Affiliations:** 1Department of Interventional Radiology, The University of Texas MD Anderson Cancer Center, Houston, TX 77030, USA; ajbarcena@mdanderson.org; 2College of Medicine, University of the Philippines Manila, Manila 1000, Philippines; 3Department of Mechanical & Aerospace Engineering, University of Texas at Arlington, Arlington, TX 76019, USA; kashish.dhal@mavs.uta.edu (K.D.); parimalthakorbh.patel@mavs.uta.edu (P.P.); 4Department of Radiology, University of Cincinnati, Cincinnati, OH 45219, USA; raviph@ucmail.uc.edu; 5Department of Biostatistics, Division of Basic Science Research, The University of Texas MD Anderson Cancer Center, Houston, TX 77030, USA; skundu2@mdanderson.org; 6Department of Breast Imaging, Division of Diagnostic Imaging, The University of Texas MD Anderson Cancer Center, Houston, TX 77030, USA

**Keywords:** 3D printing, additive manufacturing, biomedical engineering, hydrogels, polymers, tissue engineering and regenerative medicine

## Abstract

Three-dimensional (3D) printing, also known as additive manufacturing, has revolutionized the production of physical 3D objects by transforming computer-aided design models into layered structures, eliminating the need for traditional molding or machining techniques. In recent years, hydrogels have emerged as an ideal 3D printing feedstock material for the fabrication of hydrated constructs that replicate the extracellular matrix found in endogenous tissues. Hydrogels have seen significant advancements since their first use as contact lenses in the biomedical field. These advancements have led to the development of complex 3D-printed structures that include a wide variety of organic and inorganic materials, cells, and bioactive substances. The most commonly used 3D printing techniques to fabricate hydrogel scaffolds are material extrusion, material jetting, and vat photopolymerization, but novel methods that can enhance the resolution and structural complexity of printed constructs have also emerged. The biomedical applications of hydrogels can be broadly classified into four categories—tissue engineering and regenerative medicine, 3D cell culture and disease modeling, drug screening and toxicity testing, and novel devices and drug delivery systems. Despite the recent advancements in their biomedical applications, a number of challenges still need to be addressed to maximize the use of hydrogels for 3D printing. These challenges include improving resolution and structural complexity, optimizing cell viability and function, improving cost efficiency and accessibility, and addressing ethical and regulatory concerns for clinical translation.

## 1. Brief History and Terminologies in 3D Printing

Three-dimensional (3D) printing, also known as additive manufacturing, is formally defined by the International Organization for Standardization (ISO) and the American Society for Testing and Materials (ASTM) as a process of joining materials to make parts from three-dimensional model data, usually layer upon layer, as opposed to the other fundamental manufacturing methodologies (i.e., subtractive and formative manufacturing) [1]. In subtractive manufacturing processes, the desired product is attained through the selective removal of materials (e.g., milling and drilling), while in formative manufacturing processes, the desired product is formed by subjecting materials to mechanical or restricting forces (e.g., forging and molding) [2]. The first 3D printing technology, stereolithography (SLA), was developed and patented by Charles Hull in 1986. SLA employs ultraviolet (UV) light to promote the crosslinking of light-sensitive liquid polymers in a vat, which results in the formation of solid cross-sections that adhere to each preceding layer until the final object is formed [3]. After obtaining the patent for SLA, Hull founded a company called 3D Systems Corporation. In 1989, Carl Deckard from the University of Texas at Austin patented another 3D printing technology called selective laser sintering (SLS). SLS uses lasers to selectively fuse powdered materials spread on a platform [4]. In the same year, Scott Crump filed a patent application for fused deposition modeling (FDM), a process that involves heating and extrusion of thermoplastic materials to form objects [5]. Scott Crump and his wife, Lisa Crump, co-founded a company called Stratasys. In 1993, Emanuel Sachs, John Haggerty, Michael Cima, and Paul Williams from the Massachusetts Institute of Technology patented a technology called “Three-dimensional printing techniques”, which gave rise to the popular term we now know as 3D printing [6]. Since then, the art and science of 3D printing have evolved. Stratasys and 3D Systems Corporation are now the world’s two largest 3D printing companies, and numerous new technologies have been developed over the recent decades.

The expansion of 3D printing technologies has been accompanied by an increase in the variety of names used to describe each of them. It is worth noting that some of these phrases are exclusive trademarks, while others are non-standard or lack clear definitions. The establishment of a standardized language is crucial within the realm of clinical research and medical practice due to its role in fostering clarity and reproducibility [7]. The foremost standard for 3D printing is the ISO/ASTM 52900, which classifies 3D printing technologies into seven process categories [1]. The Radiological Society of North America has also integrated these standard 3D printing terminologies into the RadLex project, a comprehensive set of radiology terms for use in radiology reporting, decision support, data mining, data registries, education, and research [8]. These standard categories, along with associated commercial terms, subtypes, and RadLex identifiers, are shown in Table 1.

## 2. Hydrogels and 3D Printing

Hydrogels are hydrophilic polymeric three-dimensional networks that possess the ability to absorb and retain a substantial quantity of water. This unique property is attributed to either physical or chemical crosslinking of individual polymer chains within the hydrogel [9]. Later, this property was discovered to possess significant potential for many biological applications. Various 3D printing techniques and hydrogel compositions are currently employed to efficiently and precisely produce complex biomimetic structures.

### 2.1. Emergence of Hydrogels as a 3D Printing Feedstock Material

The term hydrogel was coined in 1894, but it was used to describe a colloidal gel as opposed to hydrophilic polymeric networks [10]. In 1960, Wichterle and Lim reported the first biological application of polymeric hydrogels [11]. During that period, the use of polymers in the production of prostheses was limited due to concerns about their biocompatibility and physicochemical properties. There was a growing need for synthetic materials that could replace or augment human tissue structure and function, but most synthetic polymers are rigid and are difficult to integrate into soft tissues. Hence, Wichterle and Lim developed a polymeric gel with hydrophilic groups that allowed the retention of water and consequently optimized the hardness and biocompatibility. Due to their substantial water content, hydrogels exhibit a level of flexibility that closely approaches that of natural tissue. The first hydrophilic gel, which was synthesized from poly-2-hydroxyethyl methacrylate (PHEMA), is soft enough to prevent mechanical irritation, inert to normal biological processes, and permeable to physiologic metabolites [11]. Contact lenses were one of the first successful applications of this hydrogel. To date, PHEMA continues to be a fundamental component of soft contact lenses and is currently being explored for various biomedical applications, such as bone tissue regeneration, wound healing, and cancer therapy [12]. Ultimately, the use of PHEMA in the field of ophthalmology has resulted in the emergence of a distinctive category of hydrogels referred to as biomedical hydrogels.

Hydrogels are manufactured using many conventional chemical methods. The procedures encompass a range of one-step techniques, such as polymerization and parallel crosslinking of multifunctional monomers, as well as multi-step processes that involve synthesizing polymer molecules with reactive groups and subsequently crosslinking them [13]. The crosslinking process, which facilitates sol–gel transition, may be induced via physical or chemical-based methods [14]. Chemical crosslinking results in the formation of covalent bonds between the functional groups of the materials. Examples of reactions that lead to chemically crosslinked hydrogels include 1-Ethyl-3-(3-dimethylaminopropyl)carbodiimide (EDC) coupling, Diels–Alder reaction, free radical polymerization, Huisgen 1,3-cycloaddition, Schiff base reaction, and thiol-ene addition [15,16]. On the other hand, physical crosslinking arises from non-covalent interactions, such as hydrogen bonding, hydrophobic interactions, and ionic interactions [13]. The aforementioned procedures may be optimized in order to generate a diverse range of hydrogels that possess the ability to replicate the extracellular matrix (ECM) found in different types of tissues. Hydrogels can demonstrate porous structures that are well-suited for accommodating diverse environmental conditions. Additionally, they can feature a high density of cell seeding and ensure a uniform dispersion of cells inside the scaffold [17]. Nevertheless, bulk fabrication of hydrogels is limited to producing simpler structures, such as matrices, films, or spheres [13]. In order to completely maximize their capabilities, it is essential that biomedical hydrogels be manufactured in a manner that effectively replicates the resolution and 3D architecture of various tissues and organs. Recent years have seen the emergence of hydrogels as 3D printing feedstock materials to address the need for producing complex biomimetic structures. Indeed, the advent of 3D printing technology, which allows for unparalleled control, adaptability, efficiency, and accuracy compared to traditional production methods, has facilitated novel opportunities for the production of intricate structures with complicated geometries, straight from digital designs [9,18].

### 2.2. Techniques for 3D Printing Hydrogels

Hydrogels are comparatively softer than other commonly used polymers in 3D printing, and consequently, need gentler processing conditions. Hence, not all existing 3D printing techniques can be effectively used for the fabrication of hydrogel scaffolds. The most commonly used techniques to fabricate hydrogel scaffolds are material extrusion, material jetting, and vat photopolymerization. It is important to note that certain elements of these techniques may also be modified or combined to maximize their advantages.

Material extrusion is a process in which the material is selectively dispensed through a nozzle or orifice [1]. The primary benefits of this approach lie in its simplicity and cost-effectiveness. A diverse array of materials may be efficiently printed, and the implementation of necessary hardware and software modifications can be seamlessly executed. However, one significant limitation associated with extrusion-based printing is its restricted applicability to fluids with high viscosity despite shear thinning. In addition, nozzle blockage may hinder the appropriate formation of the desired 3D framework [9]. Overall, material extrusion allows the handling of hydrogel inks with high cell densities, but it provides lower printing speeds and resolution (100 μm) in comparison to other printing techniques primarily due to the exponential increase in required pressure drop across the nozzle for achieving higher volumetric throughput with reducing nozzle diameter [19].

Material jetting is a process in which droplets of feedstock materials, such as photopolymer resin and wax, are selectively deposited [1]. This technology facilitates the formation of heterogeneous cell constructs with precisely defined positions and offers exceptional control over the deposition pattern. Hence, higher printing resolution (50 μm) and printing rates are usually achieved with this technique [9]. However, it typically requires hydrogel inks with a lower viscosity and lower cell densities [19]. Other problems encountered with inkjet-based printing are non-uniform droplet formation, inaccuracies in the location of deposition, and loss of cell viability due to heat or electricity [9].

Vat photopolymerization is a process in which liquid photopolymers in a vat are selectively cured by light-activated polymerization [1]. In contrast to nozzle-based methods, light-assisted 3D printing has the potential to provide substantial improvements in both printing speed and resolution (20–200 μm). A diverse spectrum of viscosities is permissible, but it is essential that the hydrogel inks possess the capability of undergoing photocrosslinking. Furthermore, in order to provide optimal light penetration depth, which directly impacts the quality of the final product, it is essential that these materials possess transparency to the specific light source used [19]. The modification of polymers using methacrylic anhydride is a widely used technique to introduce photocrosslinking characteristics. Polymers that have been substituted may undergo photocrosslinking when exposed to light [20]. Interestingly, the development of hydrogel composites featuring enhanced mechanical properties and printability has extended the application of photopolymerization to polymers that are not necessarily submerged in a vat or a rigid container. For example, viscous hydrogels may be extruded to fabricate self-supporting scaffolds that can be further stabilized by photocrosslinking.

### 2.3. Hydrogel Compositions Used in 3D Printing

Nature-derived polymers, synthetic polymers, and inorganic materials have distinct and important properties and are typically selected based on the application and functional requirements of the target hydrogel. Hybrid hydrogels, often referred to as composite hydrogels, are created by the amalgamation of nature-derived and synthetic polymers, hence including the advantageous properties of both components. A wide range of materials may be used in the fabrication of hydrogels, making it an inexhaustible field of study. Therefore, this section will focus on the polymers that have been used most recently.

#### 2.3.1. Nature-Derived Polymers

Nature-derived polymers are generated from biological sources and thus often show excellent biocompatibility, biodegradability, and cell-interaction features. These polymers can be derived from a wide variety of bacteria, plants, animals, and human tissues.

The utilization of bacteria-derived polymers presents the benefit of more ecologically sustainable manufacturing processes, characterized by lower temperature requirements and reduced dependence on potentially hazardous substances in contrast to conventional chemical synthesis [21]. Most bacteria-derived polymers that have been used to 3D print hydrogels are polysaccharides, which include bacterial nanocellulose, dextran, and gellan gum. Bacterial nanocellulose is a homopolymer of β-1,4 linked glucose, produced by the fermentation of certain bacterial species. The most efficient production of bacterial nanocellulose comes from the *Acetobacter* species. It is a potential scaffold for biomedical applications due to the unique self-assembly of secreted fibrils into a nanostructured biomaterial [22]. Because of its favorable toughness and elasticity, it may serve as a mechanical enhancer, hence augmenting the mechanical strength of bioinks. Recently, it has been used to create an engineered scaffold for the treatment of full-thickness wounds in mice [23]. Dextran is another bacteria-derived polysaccharide that can be used to fabricate hydrogels. It is an exopolysaccharide composed of α-D-glucopyranose subunits with mostly α-1,6 glycosidic bonds. The extent of branching observed in dextran is contingent upon the specific strain of bacteria used, whereby enhanced linearity signifies improved water solubility [21]. It has been recently used to fabricate stem cell-loaded hydrogel bone tissue engineering [24] and a 3D culture of astrocytes [25]. Gellan gum is an extracellular polysaccharide secreted by *Sphingomonas elodea*, and it has been used to create a composite scaffold that could stimulate angiogenesis and bone regeneration [26].

Plants possess a significant abundance of hydrophilic polysaccharides, which may be obtained at a comparatively low cost of production. These polymers include cellulose [27], starch [28], and cyclodextrins [29]. Cellulose is the most abundant naturally occurring polymer of glucose and is the main component of plant cell walls. As previously mentioned, this polysaccharide is also produced by certain bacteria. In both bacterial and plant cellulose, the glucose units are held together by β-1,4 glycosidic linkages. However, bacterial and plant cellulose differ in macromolecular structures and physical properties [30]. Hydrogels may be synthesized from cellulose in its pure form by physical crosslinking, facilitated by the hydroxyl groups present. However, cellulose can undergo further modifications to finely adjust its characteristics. The majority of water-soluble cellulose derivatives are acquired by the process of etherification, where the hydroxyl groups react with organic compounds, such as methyl and ethyl groups. An example of this derivative is methylcellulose, a well-known thickener and a bulk-forming laxative, which has been recently used to optimize the 3D culture of lung cancer cells [31]. Nanostructures, namely nanofibers and nanocrystals, may also be generated from plant cellulose by use of mechanical techniques and acid hydrolysis. These nanostructures have also been explored for the 3D culture of neuroblastoma cells [27] and the fabrication of a hydrogel-based stent [32]. Starch is a polymer composed of glucose units that are bonded together by α-1,4 and α-1,6 linkages. It is abundantly found in plant-based sources, such as corn, potatoes, and wheat, and it has been used to fabricate a hydrogel scaffold for the 3D culture of fibroblasts [28]. Cyclodextrin is a cyclic oligosaccharide composed of glucopyranoside units linked through α-1,4 glycosidic bonds. It is characterized by a hydrophilic exterior and an internal cavity that can be loaded with pharmacologic agents [29]. It has been used as a component of small molecule-loaded hydrogels for the repair of spinal cord injury [33,34].

The marine ecosystem also provides a diverse range of biological resources, including several natural polymers and bioactive compounds. Hydrogels have been fabricated from chitosan, a cationic linear polysaccharide co-polymer of N-acetyl-D-glucosamine and D-glucosamine units. It is produced from the deacetylation of chitin, a polymer derived from crustacean exoskeletons and the second most abundant natural polymer following cellulose [35]. Chitosan-containing hydrogels have been used in skin tissue engineering [36] and the development of 3D cultures for astrocytes [25], lung cancer cells [31], and osteosarcoma cells [37]. Hydrogels have also been synthesized from a wide variety of seaweed-derived polymers, such as agarose, alginate, and laminarin. Of these polymers, alginate has been used most extensively due to its relatively simple gelation process. Alginate is a brown algae-derived polysaccharide composed of irregular blocks of β-D-mannuronic acid and α-L-guluronic acid residues. It forms hydrogels through ionotropic gelation with divalent cations, such as calcium, which crosslink the polymer chains to generate an “egg-box” model [38]. Alginate has been recently used to create hydrogels for tissue engineering, 3D modeling, drug screening, and drug delivery. Agarose, the main component of agar, is a polysaccharide composed of alternating β-D-galactose and 3,6-anhydro-L-galactose units. Agarose gels, due to their ability to generate stable and solid gels, have found use in biochemistry as supports for electrophoresis and protein immobilization [39]. Agarose has been used to create a hydrogel for the 3D culture of endothelial cells, fibroblasts, melanoma cells, and mesenchymal stem cells (MSCs) to study the differential response of 3D culture to chemotherapy [40]. Laminarin, a β-D-glucan produced from brown algae, is a biocompatible material that has not been extensively studied. Recently, it has been used to co-culture breast cancer cells, fibroblasts, and osteoblast precursors [41].

A wide variety of polymers are also derived from animals. These include glycoproteins and glycosaminoglycans (GAGs). Glycoproteins are protein polymers that comprise oligosaccharide chains that are covalently bonded to the side chains of amino acids. Examples of glycoproteins that have been used to create hydrogels for 3D printing include collagen, gelatin, laminin, fibrin, and fibrinogen. Collagen, the most abundant protein in the human body, is a triple-helical protein that forms fibrils and bundled fibers, which can then crosslink to produce a hydrogel matrix [42]. Its fibrous structure gives different collagen-rich tissues strength and flexibility. Hence, hydrogels made from collagen can offer a stable biomimetic environment for cell development and tissue repair. Indeed, it has been widely used in engineering tissues, such as bone [43] and skin [44], as well as in the development of 3D cultures [28,40,45] and drug-screening platforms [46,47,48]. Collagen may be broken down into smaller peptides by enzymatic or acid treatment, yielding gelatin. Although treatment destroys the triple helical structure of collagen, gelatin preserves some of the functional features of collagen, such as its biocompatibility and ability to form a hydrogel. Gelatin can be derived as a by-product of meat processing, making it a very cost-effective substitute for collagen. However, gelatin hydrogels, which are created by a process of temperature reduction, exhibit inherent instability when exposed to physiological temperatures. To overcome this limitation, side group modifications that could allow chemical crosslinking were developed. The process of replacing lysine groups in gelatin with methacryloyl groups derived from methacrylic anhydride results in the formation of gelatin methacrylate (GelMA). This modified gelatin may undergo photocrosslinking to form a hydrogel by radical polymerization facilitated by the presence of a photoinitiator when exposed to either UV or visible light [49]. GelMA, which consists of gelatin with methacrylamide or methacrylate groups, is currently one of the most widely used substances commercially available from a number of fabrication companies. Laminin is an important component of the basement membrane, and it plays an important role in cell adhesion, migration, and tissue organization. It has been used to create a hydrogel for the 3D culture of chronic lymphocytic leukemia cells [50]. Lastly, fibrinogen and its cleavage product, fibrin, can form hydrogels from the polymerization of fibrin strands, which can aggregate to yield a three-dimensional network capable of retaining a significant amount of water and biological components. These protein polymers have been used to fabricate hydrogels for bone [51], cartilage [52], muscle [53], and skin tissue engineering [54]. GAGs, such as chondroitin sulfate, dermatan sulfate, heparan sulfate, heparin, hyaluronic acid, and keratan sulfate, are linear polysaccharides composed of repeating disaccharide units made up of an amino sugar and an uronic acid. They are essential components of the ECM in a wide variety of tissues and serve essential functions in cell signaling, development, and maintenance. Moreover, the GAGs present in the ECM play a significant role in maintaining the mechanical integrity of tissues. This is attributed to their ability to store substantial quantities of water, which facilitates the hydration of ECM and enhances its resistance against compressive stresses [55]. The most commonly used GAGs for 3D printing of hydrogels are chondroitin sulfate and hyaluronic acid.

Different natural polymers may be blended in order to replicate the intricacy of native tissues. However, the successful implementation of this approach necessitates the optimization of the composite materials and the accurate replication of structures and functions found in native tissues. To address this limitation, natural composite materials have also emerged as materials for hydrogel fabrication. The most widely used natural composite is decellularized ECM (dECM), a naturally derived scaffold that is obtained from tissues or organs by the elimination of cellular elements. It maintains the 3D architecture of the original tissues and it may retain several cell growth factors, which can enhance the growth, migration, proliferation, or differentiation of seeded cells [56]. It has been used to synthesize hydrogels for 3D printing of cartilage [57,58,59,60], ovarian [61], testicular [62], pancreatic [63,64], and skin tissues [65,66,67].

#### 2.3.2. Synthetic Polymers

Synthetic polymers, which can be manufactured through simple processes and are highly customizable, are also commonly employed in the fabrication of hydrogels. Different types of synthetic polymers have varying degrees of water affinity, mechanical stability, and responsiveness to external stimuli that can be utilized for various biomedical applications.

Hydrophilic polymers are polymers that have a high affinity to water, and this property translates to solubility or swellability. Hence, they can be used to mimic the hydrated microenvironment seen in various tissues. The most commonly used hydrophilic synthetic polymers are polyethylene glycol (PEG) and polyvinyl alcohol (PVA). PEG is produced using a ring-opening polymerization of ethylene oxide to produce a broad range of molecular weights and geometries. Additionally, it is possible to activate it through the substitution of the terminal hydroxyl group with an assortment of reactive functional end groups, which would facilitate crosslinking and conjugation. Acrylate-terminated PEG can be used to achieve photopolymerization under mild conditions. PEG and polyethylene glycol diacrylate (PEGDA) have been incorporated in hydrogels for engineering biliary [68], bone [69], cartilage [70,71], nervous [72,73,74], and vascular tissues [75]. It has also been used to fabricate 3D-printed cell culture scaffolds [25,76,77] and novel delivery systems [78,79]. PVA is another hydrophilic polymer that is commonly used to fabricate hydrogels. It is synthesized through the polymerization of vinyl acetate followed by hydrolysis, which leads to the replacement of the ester group in vinyl acetate with a hydroxyl group. It has been used as a component of 3D-printed hydrogels for bone [80] and cartilage tissue [81] engineering, as well as novel drug delivery systems for retinal disease [82].

While hydrogels are characterized by their capacity to hold a significant amount of water, the incorporation of hydrophobic polymers may nonetheless play a role in their production. Hydrophobic polymers with excellent biocompatibility and durability could provide structural support to hydrogel scaffolds. Moreover, these polymers can also be tuned to encapsulate and release hydrophobic drugs and bioactive substances. The commonly used hydrophobic synthetic polymers include polycaprolactone (PCL), poly(lactic acid) (PLA), and co-polymers, such as poly(lactic-co-glycolic acid) (PLGA) and poly(lactide-co-caprolactone) (PLCL). PCL is a biodegradable semicrystalline polyester with a low melting point (55–60 °C) and a high degree of solubility in a wide variety of organic solvents. The degradation rate of PCL is notably slower compared to other commonly used biodegradable polymers, such as PLA, due to its longer aliphatic chain. It has been used as a component of 3D-printed hydrogels for engineering bone [80,83,84,85], cartilage [52,60,86], nervous tissue [73], skin [54], and tendon [87]. PLA is another biodegradable hydrophobic polyester, and it is synthesized from ring-opening polymerization of lactide constituents. Compared to PCL, it has a higher melting point (160–180 °C) and a faster degradation rate. PLCL is synthesized from the copolymerization of lactic acid and caprolactone monomers. It demonstrates a distinctive combination of characteristics originating from its individual constituents, such as mechanical and thermal stability. Recently, it has been used to fabricate a regenerative conduit for axonal regeneration [88]. PLGA is another synthetic co-polymer, and it is produced from the polymerization of lactic acid and glycolic acid. Its physicochemical properties are significantly influenced by the molar ratio at which lactic acid and glycolic acid are combined. Lactic acid exhibits greater hydrophobicity in comparison to glycolic acid and is often the predominant monomer found in co-polymer formulations. Hence, higher lactic acid content in relation to glycolic acid results in reduced rates of degradation and drug release [89]. It has been used as a component of 3D-printed hydrogels for engineering biliary [68] and cartilage tissues [90].

Stimuli-responsive polymers, a unique category of synthetic polymers, have garnered considerable interest for their capacity to facilitate the controlled release of pharmaceutical agents. Thermoresponsive polymers are a class of stimuli-responsive polymers that undergo a reversible change in their physical state in response to variations in temperature [91]. These include poly(N-isopropylacrylamide) (PNIPAM), poly(N-acryloyl glycinamide) (PNAGA), and poly(sulfobetaine methacrylate) (PSBMA). PNIPAM is a polymer that demonstrates a lower critical solution temperature (LCST) behavior in aqueous solutions. At temperatures below the LCST, it is entirely fluid, but when heated, it undergoes a phase transition and becomes insoluble. The utility of this polymer for temperature-sensitive drug release has been demonstrated in several in vitro studies [92,93,94], but it has been recently used to synthesize a hydrogel that outperforms current clinical practice in eradicating chronic methicillin-resistant *Staphylococcus aureus* orthopedic infection in an in vivo model [16]. In another in vivo animal study, it was used as a component of a 3D-printed responsive hydrogel that has been shown to enhance skin flap survival [95]. It was also recently used to fabricate a hydrogel for engineering cartilage [96]. PNAGA is another example of a thermoresponsive polymer. As opposed to PNIPAM, it exhibits an upper critical solution temperature (UCST) behavior. Hence, it becomes soluble at higher temperatures. It has been used recently to fabricate a hydrogel for cartilage tissue engineering [97]. PSBMA is another thermoresponsive polymer that has been used in cartilage tissue engineering [96]. Interestingly, it can exhibit both UCST and LCST behavior depending on the length of the substituents on the nitrogen atom [98]. Lastly, poly(3,4-ethylenedioxythiophene) (PEDOT) is a polymer that is often mixed with an anionic polymer, polystyrene sulfonate (PSS). Together, these polymers yield an intrinsically conductive polymer that may be utilized to make stimuli-responsive hydrogels for nervous tissue engineering and biosensor fabrication. It has been recently used to fabricate a hydrogel that can treat lesions in the injured spinal cord of rats [99].

#### 2.3.3. Inorganic Materials

Inorganic materials or substances or compounds that do not contain carbon-hydrogen bonds, such as ceramics, glasses, and metals, are incorporated into hydrogels through various methods to enhance their properties.

Ceramics refer to inorganic, non-metallic materials, and they can include crystalline and amorphous structures that are typically hard and chemically non-reactive. Unlike most ceramics, which are crystalline, glass is typically amorphous. The production of glass involves melting a combination of silica, soda ash, and limestone, followed by a quick cooling process that results in the formation of an amorphous structure. Bioactive ceramics are a special class of ceramic materials that have been specifically engineered to elicit targeted biological responses in order to facilitate the regeneration and repair of injured or diseased tissues and organs. In the context of bone tissue regeneration, bioactivity refers to the inherent capacity of a material to establish direct contact with live bone after its implantation in bony defects. The process by which new bone is generated on the external surfaces of bioactive ceramics is referred to as osteoconductivity [100]. The use of ceramic materials has been extensive in the fabrication of hydrogels for bone tissue engineering to reflect the biomechanical properties of natural bone. Examples of ceramic materials that have been used to provide strength, stiffness, and stability to bone scaffolds include amorphous magnesium phosphate (AMP), beta-tricalcium phosphate (β-TCP), bioactive glass, laponite, nano-attapulgite, nHA, and wesselsite. AMP has been incorporated into a 3D-printed hydrogel for bone tissue engineering, and it significantly increased bone formation after implantation [101]. β-TCP, a calcium phosphate ceramic with a chemical formula of Ca₃(PO₄)₂, is a versatile material that has been widely used in bone tissue engineering due to its biocompatibility and ability to support bone formation. Its incorporation into a hydrogel scaffold produced a construct with a compressive strength similar to that of cancellous bone and can induce bone formation without inflammation [102]. Laponite is a disc-shaped nanoparticle with a thickness of 1 nm and a diameter of 25 ± 2 nm. It belongs to a family of materials called nanoclays, which are nanoparticles derived from layered silicates. Recent studies have shown that the addition of laponite improves the printability, porosity, and osteoconductivity of 3D-printed hydrogels for bone tissue engineering [103,104]. Because of its negatively charged surface and positively charged edges, laponite can also be loaded with charged growth factors and drugs. In one study, the addition of laponite was effective in delaying the release of multiple growth factors from the hydrogel loaded with platelet-rich plasma (PRP), which promoted enhanced bone regeneration [105]. Nano-attapulgite is a nanoscale magnesium silicate mineral with rod-like crystalline morphology. The nano-rods can interact and form a high-viscosity network that can improve the printability of hydrogel inks. Indeed, it has been shown that the addition of nano-attapulgite to a 3D-printed hydrogel for bone tissue engineering has improved its printability and effectively promoted bone regeneration in a rabbit model of bone defect [106]. Hydroxyapatite, the primary mineral component of natural bone, comprises around 65% of bone weight. The composition of natural hydroxyapatite exhibits more variability due to the inclusion of minor elements such as magnesium and strontium. In contrast, synthetic hydroxyapatite exhibits higher purity levels, a well-defined chemical composition, and a significant degree of crystallinity. Nanohydroxyapatite (nHA) was developed to overcome the limitations of hydroxyapatite in bulk form, such as brittleness, low fracture resistance, and prolonged resorption. It can be synthesized through a wide variety of techniques, and it has been incorporated as a reinforcing filler, to enhance the mechanical stability of composite scaffolds and facilitate interactions with cells [107]. Furthermore, due to its small particle size and huge surface area, it rapidly resorbs and can then be replaced by natural bone within weeks [108]. Wesselsite, first discovered in the Wessels Mine in the Kalahari Manganese Field of South Africa, is a complex silicate mineral that forms micron-sized subhedral plates [109]. It has been used as a shell coating for a 3D-printed hydrogel used in bone tissue engineering, and it transformed into an interconnecting network of microchannels for bone revascularization as the hydrogel degraded in vivo [83].

Aside from ceramic materials, metallic microparticles and nanoparticles can also be integrated into hydrogel inks to confer enhanced bioactivity or controlled release capacities. Copper-based nanoparticles have been used as photothermal, photodynamic, and chemodynamic agents against cancer. Photothermal therapy involves the use of light-absorbing materials, such as metallic nanoparticles, to convert light energy into heat, which can selectively kill cancer cells. On the other hand, photodynamic therapy involves the use of photosensitizing agents that produce reactive oxygen species (ROS) in response to light of a specific wavelength. Lastly, chemodynamic therapy involves the use of specific chemical reactions, often mediated by metal ions, to induce the generation of ROS. Copper oxide nanoparticles have been incorporated into a 3D-printed hydrogel implanted in the tumor resection site to inhibit tumor recurrence. The nanoparticles served as a reservoir for releasing Cu^2+^, which produces intracellular ROS, and as a photothermal agent [110]. Likewise, copper sulfide nanoparticles have also been used to generate a hydrogel with efficient photothermal, photodynamic, and chemodynamic effects against tumors in mice [111]. Magnetic microparticles and nanoparticles have been used to develop hydrogels that can respond to magnetic fields for actuation and drug delivery. Neodymium-iron-boron microbeads have been used to create a magnetically deformable biocompatible scaffold that can be used to study the cells’ reaction to substrate deformation [112]. On the other hand, iron oxide nanoparticles have been recently incorporated into a chitosan-based hydrogel to create untethered milli-grippers that could grasp and release cargos under the influence of an applied magnetic field [113]. Likewise, iron oxide nanoparticles embedded in PEGDA have been used to fabricate the skeleton of microrobots that can respond to magnetic fields for actuation and drug delivery [78]. Metallic microparticles and nanoparticles may also be incorporated into 3D-printed hydrogels to provide responsiveness to stimuli other than magnetic fields, such as light and ultrasound. Tetrapodal zinc microparticles have been used for the adsorption and light-controlled release of vascular endothelial growth factor (VEGF) from the 3D-printed wound patch to enhance healing [114]. On the other hand, gold-nanoparticle-decorated tetragonal barium titanate has been incorporated into a hydrogel composed of GelMA and PEGDA to fabricate a piezoelectric hydrogel patch that can eliminate bacterial infection via the production of ROS under the influence of ultrasound [79].

## 3. Current Biomedical Applications

The recent biomedical applications of hydrogels can be broadly classified into four categories as shown in Figure 1—tissue engineering and regenerative medicine, 3D cell culture and disease modeling, drug screening and toxicity testing, and novel devices and drug delivery systems.

### 3.1. Tissue Engineering and Regenerative Medicine

The application of 3D-printed hydrogels has been most extensive in tissue engineering and regenerative medicine [14]. Since hydrogels can mimic the ECM of tissues, they can provide a favorable environment for the growth and differentiation of a wide variety of cell types. Through 3D printing, complex structures have been produced to mimic functional tissues and organs, such as bile ducts, blood vessels, bone, brain, cartilage, endometrium, fetal membrane, heart, kidney, larynx, liver, muscle, nerves, ovary, pancreas, skin, spinal cord, tendon, testis, and trachea.

#### 3.1.1. Bone Tissue Engineering

The skeleton is a dynamic organ that consists of specialized bone cells, mineralized and unmineralized connective tissue matrix, and vascular canals. Although it has the capacity to regenerate, several factors can limit its ability to restore its structure and function completely and efficiently. These factors include the host’s age and medical status, the defect’s size, and inflammation [115]. Three-dimensional printed hydrogels could address these limitations by directly delivering connective tissue matrix components, stem cells, and growth factors that could accelerate the healing process. Soft natural materials that have been used to create printable bone tissue include alginate, collagen, dextran, gelatin, gellan gum, fibrinogen, and hyaluronic acid, as shown in Table 2. Although prior studies have mostly restricted the use of hydrogels as fillers inside rigid scaffolds [116,117,118], the development of composite hydrogels exhibiting enhanced printability and mechanical characteristics has facilitated the use of 3D-printed hydrogels as standalone bone scaffolds [119]. To mimic the mechanical properties of natural bone, harder reinforcing materials may be added to the hydrogel bioink. These include ceramic materials, such as β-TCP, laponite, magnesium phosphate, nano-attapulgite, nHA, and wesselsite. Bioactive ceramic materials provide mechanical strength to hydrogel scaffolds, and they release bioactive ions that can promote angiogenesis and osteogenesis as they degrade [83]. Bone tissue engineering also involves the use of osteogenic cells, such as MSCs or osteoblasts, and local factors, such as bone morphogenetic proteins (BMPs) and stromal cell-derived factors (SDFs), that facilitate the growth, migration, and differentiation of osteogenic cells. In addition to augmenting the pool of cells capable of undergoing differentiation into osteoblasts, the inclusion of MSCs within the scaffold regulates inflammation to establish a microenvironment conducive to the process of bone regeneration [84]. Overall, the efficacy of 3D-printed hydrogels in bone tissue engineering has been demonstrated in several preclinical studies on mice, rats, rabbits, dogs, and pigs. These studies reveal that 3D-printed hydrogel composites with improved printability and mechanical strength can significantly improve bone regeneration by modulating inflammation and improving tissue vascularization.

#### 3.1.2. Cardiovascular Tissue Engineering

Hydrogels hold tremendous promise for cardiac and vascular tissue engineering, providing potential solutions for the management of highly prevalent cardiovascular diseases. Three-dimensional printed hydrogels have been utilized recently to produce cardiovascular tissues. One study demonstrated that a vascular scaffold made through SLA using PEGDA can be implanted in vivo as a biocompatible and perfusable porcine arteriovenous shunt [75]. In another study, a transplanted mesh made of collagen, GelMA, cardiac fibroblasts, and cardiomyocytes demonstrated long-term graft survival, vessel formation, and stabilization. It also reduced fibrosis, increased left ventricle thickness, and enhanced cardiac function in rats with acute myocardial infarction [122].

#### 3.1.3. Cartilage Tissue Engineering

Cartilage is composed of chondrocytes embedded loosely in an ECM of protein fibers and proteoglycans. Due to its avascular nature, it is difficult to restore damaged cartilage. Similar to bone tissue engineering, 3D-printed hydrogels could enhance the limited intrinsic healing capacity of cartilage tissue by allowing the creation of constructs that replicate the complex structure and organization of native cartilage [123,124]. Both natural and synthetic materials have been used to fabricate cartilage-like constructs, as shown in Table 3. Natural materials that have been used in cartilage tissue engineering include alginate, collagen, chondroitin sulfate, dECM, gelatin, fibrinogen, hyaluronic acid, PRP, and silk fibroin. These materials are often used in conjunction with synthetic polymers—such as PCL, PEG, and PVA—or inorganic substances—such as magnesium oxide nanoparticles and nHA—to provide structural integrity to the engineered scaffolds and to recreate gradient structures that mimic the layered structure of native cartilage. Strategies such as cell loading or incorporation of bioactive factors have been shown to enhance tissue regeneration outcomes. Chondrocytes or MSCs are the two most used cell sources for cartilage tissue engineering. These cells could be delivered alone or in conjunction with chondrogenic factors, such as connective tissue growth factor (CTGF), growth differentiation factor 5 (GDF-5), and transforming growth factor beta (TGF-β). Cartilage tissue engineering has demonstrated promising results in small animals, such as mice, rats, and rabbits. A study on goats has shown that engineered hydrogel constructs can promote excellent articular cartilage regeneration and confer long-term chondroprotection [90]. However, more studies are necessary to confirm the efficacy of hydrogel constructs in large animals and to investigate additional methods for enhancing the benefits found in small animals.

#### 3.1.4. Genitourinary Tissue Engineering

The use of 3D-printed hydrogels in genitourinary tissue engineering has the potential to treat a number of conditions and diseases affecting organs such as the kidney, ovary, testis, endometrium, and embryonic membrane. Experiments in vivo demonstrated that an artificial capsule made of gelatin and MSCs wrapped around the kidney could reduce epithelial cell apoptosis and mitigate renal tubular structure damage in mice with acute kidney injury. MSCs exhibited robust growth and efficient distribution inside the scaffold, presenting a potential avenue for the direct and sustained delivery of stem cell treatments to the specific kidney tissue of interest [132].

The restoration of reproductive organs is another potential application of 3D printing, which is becoming an increasingly important aspect of fertility preservation. The use of hydrogels for ovarian failure has recently been explored. In a murine model, the use of a scaffold composed of dECM, gelatin, and primary ovarian cells demonstrated enhanced neoangiogenesis, enhanced proliferation of ovarian cells, and activation of survival signals as compared to the control scaffold group [61]. Hydrogels are also being explored for the reconstruction of testicular tissue. In both in vitro and in vivo mice models, a scaffold comprised of dECM and spermatogonial stem cells demonstrated significant cell attachment and biocompatibility [62].

The endometrium, which is the epithelial layer lining the uterine cavity, has a crucial function in facilitating embryo implantation and supporting the maintenance of pregnancy. Currently, there is a lack of effective therapeutic interventions for many disorders that result in the disturbance of endometrial regeneration. The implantation of a scaffold made of alginate, gelatin, and MSCs has recently been explored in rats, and it has been shown that this approach improved not only endometrial histomorphology but also endometrial receptivity functional indicators, which partially restored embryo implantation and pregnancy maintenance functions of the damaged endometrium [133]. Lastly, hydrogel printing has also been explored to address premature rupture of membranes, defined as breakage of the amniotic sac prior to delivery. In situ printing performed on a rabbit model at mid-gestation revealed that a scaffold made of GelMA and PEGDA had a favorable sealing effect for premature rupture of membranes [134].

#### 3.1.5. Hepatic and Pancreatic Tissue Engineering

Three-dimensional printed hydrogel structures have been shown to replicate the architecture and organization of liver and pancreatic tissues. This comprises the organization of hepatocytes, biliary duct structures, and vascular networks in the liver as well as the formation of islet-like structures and vasculature in the pancreas. Electroactive hydrogel scaffolds made of oxidized hyaluronic acid and chitosan have been printed in situ within partial liver resection of rats to promote cell proliferation, migration, and differentiation in vivo [135]. On the other hand, an MSC-laden dual-layer tubular scaffold made of gelatin, methacrylic anhydride, PEGDA, and PLGA has been shown to improve bile duct repair and biliary epithelial regeneration in mice after 12 weeks [68].

For pancreas tissue engineering, islet cells can be extracted from the pancreas or derived from stem cells. Encapsulation of these cells within hydrogel bioink permits the formation of functional structures. A hybrid encapsulation system made of dECM demonstrated biocompatibility in vitro and in vivo. In addition, the pancreatic islet-like aggregates in this system exhibited structural maturation and functional enhancement due to beta-cell edge intercellular interactions when tested in rats [136]. In another study, the subcutaneous transplantation of GelMA-encapsulated islets in immunocompetent mice improved streptozotocin-induced hyperglycemia symptoms without immunosuppression for 15 weeks [64]. Similarly, a more recent study showed that printed organoids composed of GelMA and mouse islet cells can sustain the activity of islet cells while enhancing their glucose sensitivity [82]. Lastly, a composite scaffold made of dECM, hyaluronic acid methacrylate, and islet cells produced increased insulin levels in diabetic mice, maintained blood glucose levels within a normal range for 90 days, and rapidly secreted insulin in response to blood glucose stimulation [63].

#### 3.1.6. Muscle and Tendon Tissue Engineering

In muscle and tendon tissue engineering, hydrogels have shown promise for regenerating damaged or injured tissues. The structure and organization of printed constructs should resemble those of native muscle and tendon tissues. In muscle tissue engineering, the scaffold design may incorporate aligned fibers or microstructures to facilitate cell alignment and muscle tissue formation. Artificial muscle tissue made of GelMA and glycidyl methacrylated hyaluronic acid implanted in the anterior tibia of rats has been shown to respond to electrical stimulation and correspond to histologically regenerated muscle tissue [137]. In another study, both short- and long-term repair results have also demonstrated the ability of a scaffold made of fibrinogen and GelMA to enhance functional skeletal muscle tissue regeneration in a rat volumetric muscle-loss model [53]. Recently, it has been demonstrated that photopolymerized GelMA constructs directly printed from a handheld 3D printer can produce significant muscle regeneration in a mouse model of volumetric muscle loss [138]. Muscles need vascularization to survive and function, and the addition of angiogenic agents could help printed constructs create functioning vascular networks. In one study, structures made of GelMA and VEGF have been demonstrated to attach to skeletal muscle and release VEGF after direct in vivo printing in mice. The process of in vivo muscle ink printing promoted functional muscle regeneration, decreased fibrosis, and improved anabolic response [139].

For tendon tissue engineering, parallel fiber bundles may duplicate the tendon’s hierarchical structure. The scaffold should allow cell adhesion, migration, and mechanical stability. The application of multilayered scaffolds composed of GelMA, methacrylated hyaluronic acid, PCL, PLGA, and MSCs enhanced the biomechanical properties of tendon-to-bone interfaces in rabbits twelve weeks after rotator cuff reconstruction surgery [87]. In a rat model of Achilles defect, the in vivo implantation of scaffolds composed of GelMA and tendon stem/progenitor cells promoted tendon regeneration and mitigated heterotopic ossification [140]. Likewise, a photopolymerized scaffold comprised of GelMA, PRP, and tendon-derived stem cells has been shown to promote the structural and functional repair of rat Achilles tendons [141].

#### 3.1.7. Nervous Tissue Engineering

In spite of the complexity and intricate organization of nervous tissues, 3D-printed hydrogels have demonstrated tremendous potential for repairing and regenerating damaged brain, nerve, and spinal cord tissues. Since the goal of engineering nervous tissue is to restore functional neural circuits and connectivity, bioink formulations must provide neuronal cells with a microenvironment conducive to their survival, growth, and functionality. Using a canine traumatic brain injury model, one study showed that brain tissue regeneration occurs more rapidly in the group with a scaffold composed of collagen, silk fibroin, MSC secretome, and basic fibroblast growth factor [142]. In another study, ECM, methacrylated hyaluronic acid, and angiogenic growth factors were used to produce a patch that can induce significant neovascularization in the brain area of rats, as confirmed by in vivo label-free photoacoustic microscopy [143].

Rat models have also been used to demonstrate the utility of printed hydrogels for nerve repair. The combination of GelMA, PEGDA, and platelets has been shown to promote peripheral nerve repair [72]. Similarly, a regenerative conduit composed of gelatin and PLCL led to successful axonal regeneration and functional recovery [88]. Nerve-like fibers made of GelMA and methacrylated hyaluronic acid have also shown outstanding functional reconstruction results from the promotion of immune modulation, angiogenesis, neurogenesis, neural relay formations, and neural circuit remodeling [144].

Lastly, the efficacy of printed hydrogels for the repair of spinal cord injury has been shown using rat models. Scaffolds engineered from gelatin, hyaluronic acid, and PEDOT:PSS have been shown to provide a healing environment around lesions in the injured spinal cord of rats [99]. Composite scaffolds made of β-cyclodextrin, gelatin, PCL, PEGDA, and oxymatrine recruited neural stem cells from the host tissue, promoted neuronal differentiation and axon extension at the lesion site, inhibited glial scar formation, and improved motor function in rats with spinal cord injury [73]. By means of immune modulation, angiogenesis, neurogenesis, neural relay formations, and neural circuit remodeling, nerve-like fibers from photopolymerized GelMA and hyaluronic acid methacrylate promoted remarkable functional reconstruction of the spinal cord [144]. A scaffold composed of GelMA, PEDOT, PEGDA, chondroitin sulfate methacrylate, tannic acid, and neural stem cells promoted the removal of glial scar tissues, regeneration of well-developed nerve fibers, and recovery of locomotor function [74].

#### 3.1.8. Respiratory Tissue Engineering

Bioprinted scaffolds have been designed to account for the specific shape and structural characteristics of the larynx and trachea. In one study, the transplanted 3D bioprinted larynx composed of GelMA, glycidyl-methacrylated hyaluronic acid, and chondrocytes maintained the airway of rabbits [145]. On the other hand, an immunomodulatory hydrogel composed of gelatin, interleukin-10, and prostaglandin-E2 has been used to improve the function of silicone for tracheal defect repair. In vivo, only 33% of rats with bare silicone implants for tracheal defect repair survived, whereas all animals with implants containing immunomodulatory hydrogels did [146]. Further research is required to advance lung tissue engineering to produce functional constructs that mimic the alveolar structure, including the presence of airway branches and blood vessels.

#### 3.1.9. Skin Tissue Engineering and Wound Healing

The skin, which is the body’s largest organ, plays several important roles, such as external defense, regulation of temperature, and production of vitamins. After severe skin damage or certain dermatological disorders, wound healing might be disrupted or lost. The primary function of tissue-engineered skin is to restore barrier function in patients with a severely compromised barrier function [147]. Skin tissue engineering bioinks should be biocompatible, promote cell viability, and replicate the natural skin ECM. Examples of natural materials used in bioinks for skin applications include alginate, chitosan, chondroitin sulfate, dECM, egg white, fibrin, gelatin, hyaluronic acid, nanocellulose, platelet lysate, PRP, and silk fibroin, as shown in Table 4. Bioactive molecules such as nitric oxide and VEGF can be incorporated into the bioink to enhance wound healing and skin regeneration. Scaffolds containing nitric oxide or VEGF have been shown to improve wound healing by promoting angiogenesis, reducing inflammation, and inhibiting apoptosis [148,149,150]. The cellular component of most tissue-engineered skin substitutes is made up of keratinocytes and fibroblasts as part of the epidermal and dermal layers, respectively. Alternatively, some scaffolds incorporate MSCs that play an important role in the proliferation phase of wound healing by populating the wound’s site and forming a provisional ECM. MSCs also help with re-epithelization, collagen synthesis, and decreasing fibrosis by secreting a variety of growth factors [147].

### 3.2. Three-dimensional Cell Culture and Disease Modeling

Three-dimensional printing of hydrogel scaffolds allows for the customization of the spatial arrangement of different cell types and the incorporation of growth factors or bioactive molecules to guide 3D cell culture growth and model disease progression [159,160]. By incorporating various cell types within the hydrogel, it is also possible to construct multicellular organ models that more accurately reflect the complexity of human organs and human diseases. Examples of cells that have been recently used to create 3D culture models include astrocytes, breast cancer cells, chronic lymphocytic leukemia cells, colorectal cancer stem cells, endothelial cells, fibroblasts, induced pluripotent stem cells (iPSCs), keratinocytes, lung cancer cells, melanoma cells, MSCs, neuroblastoma cells, osteosarcoma cells, and pancreatic cancer cells, as shown in Table 5. Although most models have been produced in vitro, the use of hydrogels has also allowed the development of disease models in vivo. For example, a bioprinted construct fabricated from alginate, gelatin, scar dECM, and fibroblasts produced a scar model that replicated both biochemical and biophysical characteristics of scar tissue for precision drug screening and evaluation [161]. In another study, a tri-layered scaffold, which was 3D printed using agarose and collagen, was embedded with melanoma cancer stem cells, endothelial cells, fibroblasts, and MSCs. The embedded cells demonstrated elevated levels of proliferation and metabolic activity, and the multicellular hydrogel supported early onset of vascularization, exhibited a different response to vemurafenib than cell cultures, and promoted tumorigenesis in murine xenotransplants [40].

### 3.3. Drug Screening and Toxicity Testing

Aside from studying the structure and function of organs, as well as disease mechanisms, hydrogel-based models can also be used to test drug responses in a controlled and realistic environment. Compared to conventional 2D cell cultures, 3D-printed hydrogel models provide a more physiologically relevant substrate for drug screening and toxicity testing because the 3D microenvironment more closely resembles the cell–cell interactions, nutrient diffusion, and tissue architecture that exist in vivo. Bioprinted hydrogel constructs can be utilized to assess the efficacy and safety of pharmaceutical agents, identify potential adverse effects, and reduce the need for animal testing. Recent studies have focused on the development of models to study cancers, such as brain cancer, breast cancer, liver cancer, lung cancer, prostate cancer, and soft tissue cancer, as shown in Table 6. Three-dimensional printed hydrogels have also been used for liver toxicity testing [165], as well as drug screening against angiogenesis [166] and respiratory infections [48].

### 3.4. Novel Devices and Drug Delivery Systems

Three-dimensional printed hydrogels have been utilized to fabricate medical devices, such as electrodes and stents. Peripheral nerve interfacing is a promising biomedical tool that can be used to record sensory or motor signals, as well as elicit specific responses in the body by stimulating the nerves. Consequently, conditions amenable to neuromodulation may also derive advantages from this technology. However, the interfacing procedure is hindered by the current electrodes, which cannot target smaller nerves. In one study, a nerve interface that can fold itself into a cuff around a small nerve was fabricated by printing a bilayer of a flexible polyurethane resin and a highly swelling sodium acrylate hydrogel. When immersed in an aqueous liquid, the hydrogel swells and folds the electrode softly around the nerve. The simple implantation and removal of an electrode, as well as its stimulation and recording capabilities, have been demonstrated on small peripheral nerves of locusts [172]. The application of 3D-printed hydrogels in the production of stents for cardiovascular disease, enteroatmospheric fistula, and esophagitis has also been demonstrated. Alginate and PLA have been used to 3D print stents that have sufficient mechanical strength, are resistant to pseudo-physiologic wall shear stress, and are non-cytotoxic against human umbilical vein endothelial cells and macrophage-like cells [173]. On the other hand, poly(acrylamide-co-acrylic acid) and cellulose nanocrystals have been recently used to fabricate a bilayer hydrogel stent for the closure of enteroatmospheric fistulas. The hydrogel’s ability to conform precisely to the curved intestine enables it to seamlessly close the fistulas and prevent intestinal fluid overflow. [32]. Lastly, it has been shown in a rat model of radiation esophagitis that hydrogels produced from esophagus dECM can be used to fabricate a hydrogel-loaded stent that can rapidly resolve radiation-induced inflammatory response [174].

Hydrogels can also be loaded with therapeutic agents and 3D printed into specific shapes to create novel medical devices and drug delivery systems. These systems can provide localized and sustained drug delivery to specific areas of the body through the controlled discharge of medications. Three-dimensional printed hydrogels have been used to deliver drugs for various diseases, such as bone defects, cancers, cardiovascular disease, cartilage defects, retinal disease, spinal cord injury, and wounds, as shown in Table 7. Examples of therapeutic agents that have been loaded into 3D-printed hydrogels include bisphosphonates, doxycycline, gemcitabine, nucleic acids (e.g., plasmids and microRNA), reduced glutathione, small molecule inhibitors, and natural products (e.g., curcumin, kartogenin, and leonurine).

An emerging application of hydrogels is incorporating responsive components designed to respond to specific stimuli, such as calcium concentration, irradiation, magnetic field, temperature, and ultrasound, as shown in Table 8. In addition to their capacity to swell, specific polymers possess an intrinsic sensitivity to changes in temperature. Examples of these temperature-responsive polymers include PNIPAM, PNAGA, and PSBMA. These polymers undergo a reversible change in their physical state in response to variations in temperature, and this property can be utilized for controlled drug release [91]. Nonetheless, the responsiveness of polymers can be further modified through the addition of multifunctional nanoparticles. Superparamagnetic iron oxide nanoparticles have been used to fabricate constructs, such as milli-grippers and microrobots, that respond to magnetic fields not only for actuation but also for drug delivery [78,113]. On the other hand, copper nanoparticles, which are responsive to near-infrared (NIR) irradiation, have been incorporated in 3D-printed hydrogels to fabricate constructs with photothermal, photodynamic, and chemodynamic properties that can be used against cancer [110,111]. In addition to exploiting the photoactive characteristics of copper nanoparticles, laser irradiation has also been used to activate hydrogel composites that can release doxorubicin for chemo-photothermal therapy [185], as well as VEGF for wound healing [95,114]. Lastly, ultrasound-controlled release has recently been explored to improve the delivery of oxygen within a 3D-printed cardiac patch [186] and to promote the sustained release of growth factors from a 3D-printed wound patch [79].

## 4. Current Challenges and Future Directions

Despite the recent progress in the biomedical field regarding the use of hydrogels for 3D printing, there are certain challenges that must be overcome in order to fully optimize their application. These challenges include improving resolution and structural complexity, optimizing cell viability and function, improving cost efficiency and accessibility, and addressing ethical and regulatory concerns for clinical translation.

### 4.1. Improving Resolution and Structural Complexity

To facilitate the fabrication of complex functional tissue and organ models, it is critical to reproduce fine features at the cellular scale while maintaining a reasonable printing volume. However, limited spatial resolution continues to impede the progress of traditional 3D bioprinting methods in replicating structurally complex tissues. In extrusion-based techniques, it is common practice to use extruded filaments with diameters above 100 μm to restrict the shear pressures exerted on cells, guaranteeing their viability. However, this limits the capability of these methods to resolve smaller features of the native microenvironment. Photopolymerization techniques, including SLA and DLP, provide superior print resolution in comparison to extrusion-based methodologies. Nevertheless, the current resolution of these methods remains within the range of several tens of microns because lateral resolution is mostly determined by the photochemistry involved in the crosslinking process, rather than the minimum size of the laser spot. On the other hand, the thickness of the layer is naturally limited by the depth to which light may penetrate [190].

High-definition (HD) 3D printing, defined by its capability to consistently produce 3D structures with feature sizes below 50 μm, is an emerging technique that aims to reproduce cellular and even subcellular features of the native microenvironment [190]. Electrowriting and multiphoton lithography (MPL) are now the most advanced HD 3D printing processes, exhibiting the highest resolutions achieved so far. Electrowriting combines elements of electrohydrodynamic jetting and extrusion. In this technique, an electrical field induces the polymer droplet at the nozzle tip to form a conical shape (i.e., Taylor cone), from which microscale fibers are printed or written toward the collector [191]. It has been used recently to print micron-sized MSC-laden filaments (5–40 μm) [192]. MPL, also known as two-photon photopolymerization, involves multiple NIR laser pulses directed towards a photosensitive material. Polymerization only takes place inside regions characterized by a sufficiently high photon density, which leads to the simultaneous absorption of two or more photons. Each of these photons carries a portion of the necessary energy to trigger the reaction between the hydrogel ink components [193]. It has been used to implement fine structural modifications (1–5 μm) within transparent cell-laden constructs [194]. The integration of these novel techniques in the fabrication of tissues, models, and delivery systems can help improve both the resolution and structural complexity of existing hydrogel scaffolds.

### 4.2. Optimizing Cell Viability and Function

During the 3D printing process, cells endure various types of stress, which may affect their post-printing cell viability and functionality. Ensuring the preservation of optimal cell viability, particularly for more susceptible cell types such as stem cells that exhibit heightened sensitivity to various stressors, is a crucial measure in guaranteeing the proper functioning of 3D-printed constructs. General strategies that can improve cell viability include the provision of an appropriate microenvironment with sufficient nutrient and gas exchange, adequate regulation of environmental factors (e.g., pH and temperature), and mitigation of damaging forces and toxic components during the printing process [195].

To provide an appropriate microenvironment for the survival and proliferation of seeded cells, numerous research studies have explored the use of natural polymers, ECM components, and growth factors to replicate the microenvironment of native tissues. In addition to providing mechanobiological signals to support cell survival and proliferation, the incorporation of these components has the potential to expedite the integration and remodeling process of the scaffolds inside host tissues. Recent studies have also explored the promotion of vascularization through the addition of pro-angiogenic growth factors and sacrificial materials that can be penetrated by newly formed blood vessels. In the case of tissue engineering, the establishment of a functional vascular network within the printed construct must be achieved to ensure adequate nutrient and gas exchange within the structure. In the case of bone tissue engineering, it has been shown that primitive vascular networks within 3D-printed hydrogels can be formed in vitro and that these could improve the vascularization of the construct once implanted in vivo [51]. Nonetheless, more studies are needed to determine the level of control needed on the cell microenvironment to drive successful tissue vascularization and host integration.

Certain variables that have the potential to induce cell damage are intrinsic to certain printing methods, and, although they may not be entirely eradicated, they may be mitigated to a certain extent. In the case of extrusion-based techniques, the diameter of extruded filaments must be adequate to facilitate the transit of cells within the nozzle to minimize the shear stress imparted onto the cells. In the case of jetting methods, the ink viscosity and the application of heat or electricity must be optimized to minimize cell damage. The absence of shear stress in some photopolymerization techniques is beneficial for cell viability. However, the use of lasers that can generate heat as well as cytotoxic photoinitiators or crosslinkers are still potential sources of cell injury [196]. The potential consequences of these concerns are exacerbated when considering HD 3D printing, as it entails a longer duration of exposure for cells to cytotoxic compounds compared to conventional photopolymerization methods. Increasing the throughput of HD 3D printing techniques, such as MPL, without compromising their resolution is an active area of study [190].

### 4.3. Improving Cost Efficiency and Accessibility

The cost effectiveness and comparatively efficient manufacturing capabilities of 3D printing make it a compelling method for the fabrication of hydrogel-based tissues and medical devices. Several computer-aided design software programs may be obtained at no cost from different sources, therefore enabling individuals with less programming expertise to use this technology and maintain affordable fabrication expenses. However, innovation is hampered by the fact that 3D bioprinting is still predominantly in the research and development phase, where barriers to widespread adoption exist. One of the primary obstacles is the high cost associated with acquiring commercial, research-grade 3D bioprinting equipment, which exhibits a wide price range spanning from USD 5000 to far above USD 1,000,000. Moreover, a considerable challenge associated with commercial 3D bioprinting systems lies in their limited adaptability for customized applications. These platforms exhibit restricted compatibility with novel biomaterials and rely on proprietary printing software and a closed hardware environment [197]. Hence, several groups have developed low-cost 3D printers that can extrude cell-loaded constructs by repurposing 3D thermoplastic printers. One group has shown that a low-cost 3D printer, such as the FlashForge Finder, may be modified into a bioprinter for a total cost of less than USD 900. The bioprinter demonstrates a high level of precision in its movement, with a travel accuracy above 35 µm in each of the three axes. To maximize accessibility and customizability, all the components for the bioprinter conversion have been provided by the authors as open-source 3D models, along with instructions for further modifying the bioprinter for other purposes [197]. Similarly, another group developed a low-cost 3D bioprinter by modifying an off-the-shelf desktop 3D printer. The 3D bioprinter is portable, customizable, and available within a price range of around EUR 150, and is thus affordable to a broad range of research laboratories and educational institutions. The authors also provided a parts list and design files as a guide for reconstructing the device [198]. Open-source initiatives are expected to encourage more collaborations for developing and sharing low-cost hardware designs, software, and experimental protocols.

### 4.4. Addressing Ethical and Regulatory Concerns toward Clinical Translation

The use of 3D-printed hydrogels in the fields of tissue engineering and regenerative medicine is anticipated to give rise to several ethical and regulatory considerations. Crucial factors to consider include ensuring the safety, effectiveness, and quality of the printed structures, alongside examination of the ethical implications associated with the printing of functioning human tissues or organs.

Ensuring the safety and efficacy of bioprinted products for clinical use is a major regulatory challenge because of the wide array of materials that are used to fabricate these constructs. It is imperative to guarantee that the products are devoid of any microbial contamination, devoid of any toxic substances, and incapable of degrading into toxic metabolites. Hence, it is critical to establish and adhere to safety and quality standards during the development of bioprinting processes, materials, and final products. The long-term efficacy and side effects of many 3D-printed tissues and organs on the human body are also not fully understood, and it is essential to conduct more large animal studies of extended duration in order to address these particular concerns. Moreover, more mechanistic studies are needed to fully elucidate the means by which 3D-printed constructs integrate with host tissues without causing immune dysregulation.

Ethical concerns may also arise regarding the use of human cells in bioprinting. Informed consent from donors and recipients must be gained, and there should be transparent communication about the source and the utilization of cells that will be printed. It is anticipated that the concerns surrounding cell therapies will also extend to 3D-printed constructs loaded with cells. Cells that are produced in vast numbers outside of their normal environment inside the human body may lose their effectiveness and pose potential risks, leading to unfavorable effects such as the development of tumors or strong immunological responses. Hence, cells that are incorporated into the hydrogels must be carefully evaluated and characterized before, during, and after the manufacturing process in order to reliably predict whether they will be safe and effective. Stem cells from adults and cord blood do not give rise to any particular ethical considerations and are utilized extensively in clinical practice and research. However, the use of human embryonic stem cells (hESCs)—defined as cells that are derived from the inner cell mass of blastocyst-stage human embryos capable of dividing without differentiating for a prolonged period in culture and are known to develop into cells and tissues of the three primary germ layers—continues to be a subject of ethical and political controversy. Nonetheless, the National Institutes of Health continue to support the conduct of responsible, scientifically worthy human stem cell research, including hESC research, to the extent permitted by law. Alternatively, somatic cells can be reprogrammed to form pluripotent stem cells, iPSCs. The genetic composition of these cell lines is congruent with that of the somatic cell donors, making them valuable for studying diseases and perhaps for allogenic transplantation purposes. These cells circumvent the contentious ethical discussions surrounding embryonic stem cell research by eliminating the use of embryos or oocytes. Moreover, due to the comparatively benign nature of obtaining somatic cells via a skin biopsy, there are fewer apprehensions over the potential hazards to donors in comparison to oocyte donation [199].

Two clinical trials have recently evaluated the effect of 3D-printed hydrogels for surgical simulation prior to the actual procedure on intraoperative and postoperative outcomes. The first trial, which started in 2017 and ended in 2023, evaluated the use of 3D models for minimally invasive partial nephrectomy (NCT03155295), while the second trial, which started in 2018 and ended in 2023, evaluated the 3D models for percutaneous nephrolithotomy (NCT03272529). The publication of the study findings is still pending. A phase 1/2 trial on 3D bioprinted collagen hydrogel scaffold encapsulating the patient’s own auricular cartilage cells was started in 2021 (NCT04399239). The construct, called AuriNovo, is a patient-specific construct designed for the surgical reconstruction of the external ear in people born with microtia. Another phase 1/2 trial was started in 2023, and it involves the creation of a personalized 3D-printed trachea loaded with nasal cavity stem cells and nasal septum cartilage cells (NCT06051747). Likewise, the latter two study’s findings have not yet been published. The conduct of these recent trials indicates that although there are legitimate ethical and regulatory concerns regarding 3D-printed cell-laden hydrogels, the potential benefits could still outweigh the potential risks. In the coming years, 3D printing technologies will continue to expand, and as the field’s current challenges are resolved, it is anticipated that more studies and clinical trials will ensue.

## Figures and Tables

**Figure 1 gels-10-00008-f001:**
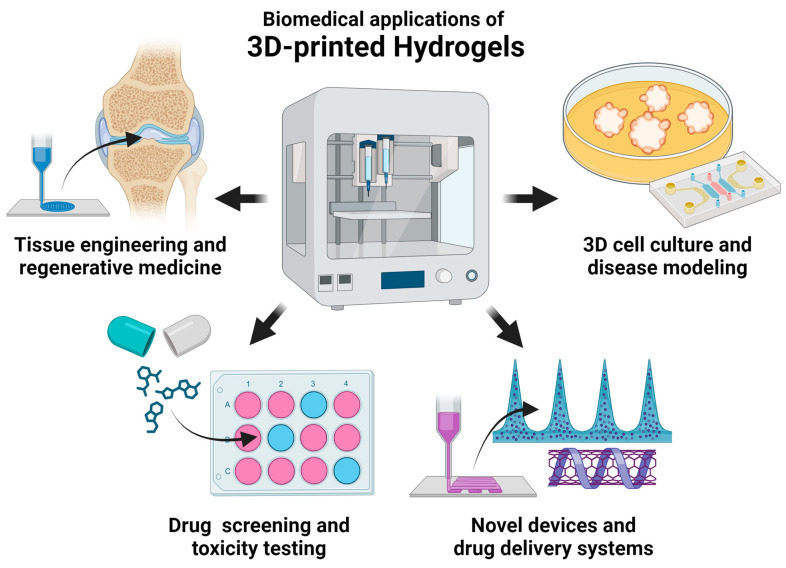
Current biomedical applications of 3D-printed hydrogels.

**Table 1 gels-10-00008-t001:** Terminologies for 3D printing technologies. Adapted with permission from Alexander et al. 2021 [7].

Generalized Standard Term	Commercial and Other Term Examples	Description	RadLex Identifier
Binder Jetting (BJT)	ProJet Color Jet Printing (CJP)	Liquid materials are selectively dropped onto powder media. Subsequent infiltration or heating may be needed.	RID50562
Directed Energy Deposition (DED)	Laser Engineered Net Shape (LENS) Electron Beam Additive Manufacture (EBAM)	Focused application of energy and material selectively melted and fused on a build platform or part.	RID50563
Material Extrusion (MEX)	Fused Deposition Modeling (FDM) Fused Filament Fabrication (FFF)	Material is dispensed onto a build platform, typically through a heated nozzle.	RID50564
Material Jetting (MJT)	Nanoparticle Jetting (NPJ) Drop-On-Demand (DOD) PolyJet ProJet Multijet Printing (MJP)	A print head dispenses droplets of material onto a build platform where each layer is solidified.	RID50565
Powder Bed Fusion (PBF)	Selective Laser Sintering (SLS) Selective Laser Melting (SLM) Direct Metal Printing (DMP) Direct Metal Laser Sintering (DMLS) Electron Beam Melting (EBM) Multi Jet Fusion (MJF)	Powder media is deposited on a build platform and subsequently bonded together through a heating process.	RID50566
Sheet Lamination (SL)	Laminated Object Manufacturing (LOM)	Discrete layers of material are fused together to form a product.	RID50567
Vat Photopolymerization (VP)	Stereolithography apparatus (SLA) Digital Light Processing (DLP) Continuous liquid interface production (CLIP)	Liquid photopolymer is selectively exposed to a light source to facilitate layer-by-layer solidification.	RID50568

**Table 2 gels-10-00008-t002:** Recent applications of 3D-printed hydrogels in bone tissue engineering.

Hydrogel Composition	Technique	Model	Physicochemical Properties	In Vitro Efficacy	In Vivo Efficacy	Ref.
Alginate, gelatin, laponite	Extrusion	Rat	Pore size: 400 μm Swelling ratio: ~10 Compressive modulus: ~65 kPa	The scaffold enhanced the proliferation and osteogenic differentiation of MSCs.	After 8 weeks, the ectopically implanted porous hydrogel improved in vivo mineralization and osteogenesis (Bone volume/tissue volume (BV/TV): 15%).	[103]
Alginate, gelatin, nano-attapulgite	Extrusion	Rabbit	Pore size: ~500 μm Swelling ratio: 3–4 Compressive strength: ~25 MPa	The scaffold supported the proliferation and enhanced the osteogenic differentiation of MSCs.	Histological analysis of tibia bone defects after 12 weeks demonstrated that the composite hydrogels effectively promoted bone regeneration.	[106]
Alginate, gelatin, laponite, MSCs	Extrusion	Rat	Pore size: ~500 μm Swelling ratio: 9.6 Compressive modulus: ~100 kPa	The scaffold supported MSC growth and enhanced osteogenic differentiation and mineralization.	Compared to scaffolds without laponite or cells, the nanocomposite scaffolds significantly accelerated bone regeneration in rat calvarial defects over 12 weeks (BV/TV: 29.82%).	[104]
Alginate, gelatin, PCL, wesselsite, SDF-1α	Extrusion	Rat	Porosity: ~65% Compressive strength: 1.07 MPa Release: 70–90% of loaded SDF-1α after 14 days	Wesselsite improved the proliferation and osteogenic differentiation of MSCs. SDF-1α improved the migration and tube formation of human umbilical vein endothelial cells (HUVEC).	After 12 weeks, the prepared scaffolds demonstrated enhanced bone repair capacity (BV/TV: 29.82%) with profuse new bone formation and blood vessel ingrowth in the region of cranial defect.	[83]
Alginate, PEGDA, GelMA	Photopolymerization	Pig	Compressive modulus: 78.1 kPa	The scaffold enhanced the osteogenic differentiation of MC3T3-E1 preosteoblast cells.	Through robotic in situ 3D printing, long segmental defects on the right tibia of pigs were repaired with significantly decreased operative time. After 3 months, the printed scaffold produced thick cortical bone tissues (BV/TV: 74.8%) with a smooth surface.	[69]
Alginate, gelatin, autologous bone, PCL, MSCs	Extrusion	Dog	Pore size: ~500 μm	The scaffold supported the survival and enhanced the osteogenic differentiation of MSCs.	The scaffold was implanted in the cranial defects of beagle dogs for up to 9 months, and it enhanced the formation of new bone (BV/TV: 17%) through the in situ differentiation of transplanted MSCs and the recruitment of native MSCs.	[84]
Alginate methacrylate, GelMA, PRP, laponite	Photopolymerization	Rat	Porosity: ~80% Swelling ratio: 0.16 Compressive modulus: 180.55 kPa Release: 90% release of growth factors over 14 days.	The addition of PRP and laponite enhanced the proliferation and osteogenic differentiation of MSCs. PRP and laponite improved in vitro HUVEC proliferation and tubule generation.	The scaffolds promoted vascular inward growth and enhanced bone regeneration after 4 weeks (BV/TV: ~27%).	[105]
Collagen, GelMA, hyaluronic acid, vinyl-modified nHA	Photopolymerization	Rabbit	Pore size: ~700 μm Swelling ratio: 3.5 Compressive strength: 20 MPa	The scaffold enhanced MSC proliferation and osteogenic differentiation.	The scaffold achieved significant bone reconstruction in the rabbit cranial defect model, obtaining 61.3% breaking load strength and 73.1% bone volume fractions in comparison to natural cranium.	[43]
Dextran, GelMA, MSCs	Photopolymerization	Rat	Pore size: 10–50 μm Compressive modulus: 0.5 kPa	The void-forming scaffold promoted the migration, proliferation, cell spreading, and osteogenic differentiation of encapsulated MSCs.	In vivo evaluations revealed that the void-forming hydrogel has the potential to deliver MSCs and can substantially promote bone regeneration in cranial defects (BV/TV: ~65%).	[24]
Fibrinogen, gelatin, HUVECs, MSCs	Extrusion	Rat	Spreading ratio: 1.76	The scaffold supported the proliferation of MSCs and enhanced the in vitro formation of a stable primitive vascular network.	Establishing microvessels within bioprinted tissues in vitro prior to their implantation resulted in improved vascularization and bone formation in femoral defects (BV/TV: ~10%) after 12 weeks.	[51]
Gelatin, β-TCP	Extrusion	Rat	Pore size: 500 μm Compressive strength: 11.45 MPa	The scaffold enhanced the proliferation and osteogenic differentiation of MC3T3-E1 cells.	The scaffold induced bone formation in calvarial defects (BV/TV: ~55%) without any inflammatory responses	[102]
Gelatin, hyaluronic acid, hydroxyapatite/PCL nanoparticles, PVA	Extrusion	Rabbit	Pore size: 71.5–116.6 μm Compressive strength: 80.1–147 kPa	Not examined	In a rabbit tibial model, the scaffold enabled osteoconduction and bone healing by serving as a template for new bone formation over 6 weeks (BV/TV: ~100%).	[80]
GelMA, gellan gum methacrylate, deferoxamine-loaded ethosomes	Photopolymerization	Rat	Swelling ratio: 3–5 Compressive strength: 282.71 kPa Release: 60% of deferoxamine over ~500 h	Deferoxamine enhanced the migration of HUVECs and osteogenic differentiation of MSCs.	By activating the hypoxia-inducible factor 1-alpha signaling pathway, the composite scaffold stimulated angiogenesis and bone regeneration in cranial defects after 12 weeks (BV/TV: 42.32%).	[26]
GelMA, MSCs	Photopolymerization	Mouse	Aligned microstructure	The aligned microstructure promoted the migration and angiogenesis of co-cultured cells and promoted the osteogenic differentiation of MSCs.	Experiments in vivo reveal that the aligned biomimetic periosteum can actively promote local angiogenesis and osteogenesis in cranial defects after 12 weeks (BV/TV: ~20%).	[120]
GelMA, MSCs	Photopolymerization	Rat	Pore size: ~500 μm	The scaffold promoted the proliferation and osteogenic differentiation of encapsulated MSCs.	In vivo implantation in a rat condyle defect model for 8 weeks revealed tissue integration and no indications of fibrotic encapsulation or bone formation inhibition.	[121]
Octapeptide hydrogel, AMP	Extrusion	Rat	Pore size: 500–1000 μm	The scaffold enhanced the osteogenic differentiation of dental pulp stem cells.	The presence of AMP in the bioink significantly increased bone formation after 8 weeks (BV/TV: ~15%).	[101]

**Table 3 gels-10-00008-t003:** Recent applications of 3D-printed hydrogels in cartilage tissue engineering.

Hydrogel Composition	Technique	Model	Physicochemical Properties	In Vitro Efficacy	In Vivo Efficacy	Ref.
Acrylamide, alginate, nHA, MSCs	Photopolymerization	Rat	Pore size: 500–1000 μm Swelling ratio: 6 Compressive strength: 900 kPa	The scaffold supported the proliferation of goat temporomandibular joint disc cells.	After 12 weeks, the MSC-loaded gradient scaffolds exhibited superior coverage of knee cartilage defect (International Cartilage Repair Society [ICRS] score: 10.67) compared to other scaffolds.	[125]
Chondroitin sulfate methacrylate, GelMA, hyaluronic acid methacrylate, TGF-β1	Photopolymerization	Rat	Swelling ratio: 8.4 Compressive strength: 82.3 kPa Release: 70% of loaded TGF-β1 after 21 days	TGF-β1 enhanced the proliferation and chondrogenic differentiation of MSCs.	After 12 weeks, the scaffold effectively promoted knee cartilage regeneration (ICRS: ~10) and functional recovery of injured joints.	[126]
dECM methacrylate	Photopolymerization	Mouse	Pore size: 50–100 μm Swelling ratio: 4 Compressive strength: ~70 kPa	The scaffold supported the proliferation of chondrocytes.	After 4 weeks, the subcutaneously implanted scaffold demonstrated cartilage regeneration and maturation.	[57]
dECM, GelMA	Photopolymerization	Mouse	Swelling ratio: 8 Compressive modulus: ~350 kPa	The addition of dECM enhanced chondrocyte viability and ECM secretion.	After 12 weeks, the subcutaneously implanted scaffold enhanced cartilage regeneration.	[58]
dECM, GDF-5	Extrusion	Rabbit	Pore size: 16.2 μm Porosity: 73.8% Compressive modulus: 97 kPa Release: 85% of loaded GDF-5 after 1 month	GDF-5 enhanced the migration and chondrogenic differentiation of MSCs.	The scaffolds recruited MSCs and provided an ideal regenerative microenvironment for them. After 12 weeks, the scaffolds significantly enhanced in situ knee cartilage repair (ICRS: ~10).	[59]
dECM, PCL, magnesium oxide nanoparticles coated with polydopamine	Extrusion	Rat	Compressive strength: 0.43–0.58 MPa Release: 30 mM of Mg^2+^ over 12 weeks	The bilayer scaffold promoted the proliferation and enhanced the chondrogenic/osteogenic differentiation of MSCs.	After implantation into a rat’s osteochondral defect, the integrated bilayer scaffold demonstrated simultaneous regeneration of knee cartilage (ICRS: 11) and subchondral bone.	[60]
Fibrinogen, gelatin, hyaluronic acid, PCL, MSCs	Extrusion	Rabbit	Pore size: 150–750 μm	Pore size-dependent chondrogenic differentiation and ECM formation were demonstrated in vitro.	The cartilage scaffold with a gradient structure demonstrated a superior knee cartilage repair effect. Blood vessel ingrowth and cartilage tissue maturation were mediated by a pore-size-dependent mechanism.	[52]
GelMA, PCL, chondrocytes, MSCs, TGF-β3	Photopolymerization	Rat	Pore size: 308.7 μm Porosity: 61.6% Compressive modulus: 7.24 MPa	TGF-β3 enhanced the proliferation and ECM deposition of MSCs and chondrocytes.	When implanted for 12 weeks in a rat model of knee osteochondral defect, the pre-cultured scaffolds demonstrated excellent cartilage regenerative capability (ICRS: ~11). The scaffold also resulted in less pain and normalization of gait.	[86]
GelMA, glycidyl-methacrylated hyaluronic acid, MSCs	Photopolymerization	Mouse	Compressive strength: ~0.7 MPa	The scaffold enhanced the proliferation and chondrogenic differentiation of MSCs.	The histological findings demonstrated that the construct’s cells survived in the subcutaneously implanted scaffold until the third week after transplantation and that cartilage-like tissues developed over time.	[127]
GelMA, MSCs	Photopolymerization	Rabbit	Pore size: 176 μm Compressive modulus: ~4 kPa	The scaffold promoted the proliferation and chondrogenic/osteogenic differentiation of MSCs.	After 12 weeks, the scaffold stimulated the regeneration of cartilage in a model of rabbit knee cartilage injury.	[128]
GelMA, PRP	Photopolymerization	Rabbit	Pore size: 127 μm Porosity: 75% Swelling ratio: 8.9 Compressive modulus: 174 kPa	PRP enhanced the proliferation, migration, and chondrogenic/osteogenic differentiation of MSCs. It also promoted M2 macrophage polarization.	After 18 weeks, the 3D-printed composite scaffold promoted osteochondral repair of knee defects (ICRS: ~11) by regulating the immune system via M2 polarization.	[129]
Gelatin, hyaluronic acid methacrylate, norbornene-grafted hyaluronic acid	Photopolymerization	Mouse	Compressive strength: ~0.21 MPa	The scaffold enhanced the proliferation and ECM deposition of chondrocytes.	After implantation for 8 weeks, in vitro-regenerated cartilage formed homogenous and mature cartilage similar to the native cartilage after subcutaneous implantation.	[130]
Glycidyl-methacrylated silk fibroin	Photopolymerization	Rabbit	Not evaluated	The scaffold supported the proliferation and ECM deposition of chondrocytes.	Experiments on a rabbit model with a partial defect in the trachea demonstrated the presence of cartilage-like tissue and epithelium surrounding the transplanted hydrogel after 6 weeks.	[131]
Hyaluronic acid methacrylate, PEG	Photopolymerization	Rabbit	Compressive modulus: 6.91 GPa	Not evaluated	The knee osteochondral defect could be repaired in approximately sixty seconds, and the regenerated cartilage in the hydrogel implantation and in situ 3D printing groups exhibited identical biomechanical and biochemical performance after 12 weeks (ICRS: ~9).	[70]
Phenylboronic acid grafted-hyaluronic acid, PVA	Extrusion	Mouse	Pore size: ~50 μm Swelling ratio: 0.5 Compressive strength: 15.5 kPa	The scaffold supported the growth and chondrogenic differentiation of MSCs.	Injecting the hydrogel intra-articularly into mice revealed stability and biocompatibility in vivo after 3 weeks.	[81]
PEGDA	Photopolymerization	Rabbit	Pore size: 250–1000 μm Compressive modulus: 1.09 MPa	The scaffold supported the growth and chondrogenic differentiation of MSCs.	After 8 weeks, the printed growth plate resulted in greater tibial lengthening than the control group but did not demonstrate cartilage regeneration in vivo.	[71]
PLGA, MSCs, CTGF, TGF-β3	Extrusion	Goat	Tensile strength: ~23 MPa	The scaffold supported the growth and enhanced the chondrogenic differentiation of MSCs.	After 24 weeks, the meniscus construct exhibited similarity to the native meniscus and conferred better mobility in daily movement.	[90]
PNAGA, poly(N-acryloylsemicarbazide)	Photopolymerization	Rabbit	Compressive modulus: 2.11 MPa	Not evaluated	A biomimetic meniscus substitute was fabricated that exhibited anisotropic mechanics comparable to those of the native tissue. At 12 weeks, the scaffold alleviated the wear of articular cartilage.	[97]
PNIPAM, PSBMA, MSCs	Photopolymerization	Rabbit	Pore size: 200 μm Porosity: 85%	The scaffold supported the growth of MSCs.	The granular hydrogel allowed the formation of numerous stem cell spheroids within the scaffold. Histological analysis revealed that the cartilage defect filled with the hydrogel was replaced with cartilage-like neotissue.	[96]

**Table 4 gels-10-00008-t004:** Recent applications of 3D-printed hydrogels in skin tissue engineering and wound healing.

Hydrogel Composition	Technique	Model	Physicochemical Properties	In Vitro Efficacy	In Vivo Efficacy	Ref.
Alginate, chondroitin sulfate methacrylate, VEGF	Photopolymerization	Mouse	Pore size: ~200 μm Release: ~90% of VEGF over 9 days.	The scaffold supported the survival of dermal fibroblasts and enhanced the migration and tube formation of HUVECs.	The scaffold produced the largest vascular area compared with the other groups, and it improved wound healing in mice with type 1 diabetes after 9 days (Wound closure [WC]: ~95%).	[148]
Alginate, gelatin, fibroblasts	Extrusion	Rat	Pore size: ~5 mm Swelling ratio: 0.8 Tensile modulus: 0.5 MPa	The scaffold supported the proliferation of dermal fibroblasts.	The scaffold enhanced the healing of deep partial-thickness burns in rats after 28 days (WC: ~95%).	[151]
Alginate, gelatin, MSCs	Extrusion	Mouse	Pore size: 32.6–103.8 μm Swelling ratio: ~2.5 Tensile modulus: 264 kPa Compressive modulus: 187 kPa	The scaffold supported the proliferation and enhanced the paracrine secretion of MSCs.	Enhanced paracrine secretion of adipose-derived stem cells enhanced angiogenesis and healing of full-thickness wounds after 14 days (WC: ~95%).	[152]
Alginate, gelatin, MSCs, nitric oxide	Extrusion	Mouse	Pore size: ~1 mm Release: ~80% of nitric oxide over 5 days	The scaffold enhanced the migration and angiogenesis of HUVECs.	The scaffold accelerated the healing of burn wounds (Wound closure: ~90%) by increasing neovascularization, epithelialization, and collagen deposition after 14 days.	[149]
Alginate, gelatin, PRP	Extrusion	Rat	Tensile strength: ~47 kPa Release: 50% burst release of PRP within 4 h	The scaffold enhanced the proliferation, migration, and function of dermal fibroblasts.	After 21 days, the integration of PRP accelerated the closure of a full-thickness defect (WC: ~95%), modulated inflammation, and initiated angiogenesis.	[153]
Chitosan methacrylate	Photopolymerization	Rat	Pore size: 1–2 mm Swelling ratio: 1–2	The scaffold supported the survival of NIH/3T3 mouse fibroblast cells.	After 21 days, the scaffolds promoted wound healing (WC: ~95%) and did not cause any adverse effects.	[36]
Collagen, platelet lysate, fibroblasts	Extrusion	Rat and pig	Pore size: 600 μm	The addition of platelet lysate improved the migration of HUVEC spheroids	In all animals, the defects completely healed within 4 weeks.	[44]
dECM, GelMA, fibroblasts, HUVECs, and keratinocytes	Photopolymerization	Mouse	Pore size: 61–196 μm Swelling ratio: 1–6 Young’s modulus: 0.61–160 kPa	The scaffold supported the proliferation of fibroblasts, HUVECs, and keratinocytes.	The multi-layered scaffold could maintain cell viability for at least one week in vivo. It stimulated dermal ECM secretion, angiogenesis, and wound healing after 14 days (WC: ~100%).	[65]
dECM, GelMA, hyaluronic acid methacrylate, MSCs	Photopolymerization	Mouse	Pore size: 73 μm Porosity: 65%	The scaffold supported the growth of MSCs.	The 3D-printed skin substitutes accelerated the healing of full-thickness wounds over 14 days (WC: ~90%).	[66]
Decellularized small intestinal submucosa, mesoporous bioactive glass, exosomes	Extrusion	Rat	Pore size: 50–500 μm Release: ~80% of exosomes over 14 days.	The scaffold enhanced the proliferation and angiogenesis of HUVECs.	The scaffolds increased wound blood flow and stimulated angiogenesis, thereby accelerating wound healing in rats with diabetes over 14 days (WC: ~90%).	[67]
Egg white	Extrusion	Mouse	Pore size: 83 μm Tensile modulus: 17.7 kPa	The scaffold promoted the proliferation and migration of fibroblasts.	In the absence of exogenous growth factors, the scaffold enhanced angiogenesis, collagen deposition, and healing of normal (WC: ~100% after 14 days) and diabetic wounds (WC: ~100% after 18 days).	[154]
Gelatin, fibrin, PCL, fibroblasts, HUVECs	Extrusion	Mouse	Displacement: 38.6 μm	The scaffold supported the proliferation of fibroblasts and HUVECs.	Perfusion was observed within the dermally implanted scaffold after 14 days. The mature vessels expanded in their original orientation with few branches.	[54]
Gelatin, polyurethane, endothelial progenitor cells, fibroblasts, keratinocytes	Extrusion	Rat	Swelling ratio: 2.91	The scaffold supported the proliferation of fibroblasts and keratinocytes.	The large and irregular rat skin wounds treated with the hydrogel demonstrated full repair after 28 days (WC: 100%).	[155]
GelMA, nanocellulose, fibroblasts, keratinocytes	Photopolymerization	Mouse	Pore size: 30–80 μm Swelling ratio: 5–9 Compressive modulus: 20–70 kPa	The scaffold supported the proliferation of fibroblasts and keratinocytes.	After 14 days, the scaffold enhanced full-thickness wound healing (WC: ~95%). The scaffold generated hair follicles and early-stage rete ridge structures, which resembled normal skin in vivo.	[23]
GelMA, silk fibroin	Photopolymerization	Mouse	Pore size: 100.54 μm Swelling ratio: 10.96	The scaffold supported the migration and proliferation of fibroblasts.	The scaffold accelerated wound healing in mice over 12 days (WC: 100%).	[156]
GelMA, VEGF	Photopolymerization	Pig	Pore size: 1 mm Swelling ratio: 2 Tensile strength: 175 kPa Release: ~85% of VEGF over 6 days	The scaffold enhanced NIH/3T3 proliferation and HUVEC tubule formation.	After 28 days, the patch accelerated wound healing by promoting collagen deposition and angiogenesis (WC: ~95%).	[150]
Hyaluronic acid methacrylate, pluronic F127, MSCs	Photopolymerization	Mouse	Pore size: 140.11 μm Swelling ratio: 7.48 Compressive modulus: 24.05 kPa	The scaffold enhanced the proliferation of MSCs.	By modulating inflammation and accelerating collagen deposition and angiogenesis, the scaffold promoted the healing of full-thickness wounds after 14 days (WC: ~100%).	[157]
Matrigel, epidermal stem cells, skin-derived precursors	Extrusion	Mouse	Not evaluated	Not evaluated	Four weeks after in situ bioprinting, the scaffolds showed successful regeneration of hair follicles and other cutaneous appendages.	[158]

**Table 5 gels-10-00008-t005:** Recent applications of 3D-printed hydrogels in skin 3D cell culture and disease modeling.

Cells	Hydrogel Composition	Technique	Cell Density	In Vitro Results	Ref.
Astrocytes	Gelatin, GelMA	Photopolymerization	2 × 10^6^ cells/mL	The scaffolds supported the 3D culture of primary rat astrocytes for 7 days. The astrocytes were homogeneously distributed within the construct and exhibited a characteristic stellar morphology.	[25]
Breast cancer cells, fibroblasts	GelMA	Photopolymerization	2 × 10^6^–1 × 10^7^ cells/mL	The scaffold supported the proliferation of MCF-7 human breast cancer cells for 14 days. Printed structures include homogeneous multi-spheroid beads and heterogeneous tumor models containing MCF-7 cells and normal human dermal fibroblasts.	[162]
Breast cancer cells, endothelial cells, fibroblasts	Collagen, hyaluronic acid, PNIPAM	Extrusion	1 × 10^5^–4 × 10^6^ cells/mL	The scaffold supported the proliferation of breast tumors for 7 days. The scaffold enhanced the formation of acinar colonies by 21PT human breast cancer cells. C3(1)-tag tumor organoids within the scaffold replicated the morphology of in vivo tumors. The co-culture of 21PT cells, H16NF fibroblasts, and HUVECs allowed the modeling of the effect of hypoxia on tumor vascularization, epithelial–mesenchymal transition, and tumor invasion.	[45]
Breast cancer cells, fibroblasts, osteoblast precursors	Alginate, boronic acid-functionalized laminarin	Extrusion	1 × 10^6^–5 × 10^6^ cells/mL	The bioprinted hydrogels allowed homogenous cell distribution and supported the proliferation of MC3T3-E1, MDA-MB-231 breast adenocarcinoma cells, and L929 fibroblasts for 14 days.	[41]
Chronic lymphocytic leukemia cells	Laminin, arginine-glycine-aspartic acid (RGD)	Extrusion	1 × 10^8^ cells/mL	The scaffold supported the proliferation of MEC1 chronic B-cell leukemia cells, and the cells showed a different gene expression profile than 2D-cultured cells. The scaffold also supported the proliferation of primary chronic lymphocytic leukemia cells for 28 days.	[50]
Colorectal cancer stem cells	GelMA, laponite	Photopolymerization	1 × 10^6^ cells/mL	The scaffolds enhanced the proliferation and sphere formation of SW480 human colon adenocarcinoma cells and hCC001 human primary cancer cells. The scaffolds also enhanced the stemness and in vivo tumorigenic potential of SW480 cells.	[98]
Fibroblasts	Collagen, gelatin, starch	Extrusion	1 × 10^6^ cells/mL	The nanocomposite starch hydrogel scaffold enhanced the proliferation rate of NIH/3T3 fibroblasts for 7 days.	[28]
Fibroblasts, keratinocytes	dECM, hyaluronic acid	Extrusion	~1 × 10^5^ cells/mL	The scaffold enhanced the proliferation of NIH/3T3 fibroblasts for 7 days. The scaffold also facilitated the co-culture of human dermal fibroblasts and HaCat human epidermal keratinocytes. An artificial skin was created through the implantation of human dermal fibroblasts and keratinocytes into 3D-printed hydrogels.	[163]
Fibroblasts, melanoma cells	GelMA, PEGDA	Photopolymerization	2 × 10^6^ cells/mL	The multicellular 3D model composed of A375 human melanoma cells and human fibroblasts displayed higher proliferation than the A375 model over 7 days. The multicellular culture model also had higher MMP-2, higher MMP-9, lower E-cadherin, higher VEGF expression, and higher resistance to luteolin.	[76]
iPSCs	h9e peptide	Extrusion	2 × 10^5^ cells/mL	The scaffolds supported the proliferation and spheroid formation of iPSCs. The spheroids in the peptide hydrogel exhibited superior pluripotency and differentiation potential based on multiple biomarkers.	[164]
Lung cancer cells	Agar, alginate, chitosan, gelatin, methylcellulose	Extrusion	Not reported	The bioinks supported the proliferation of H69AR epithelial lung cancer cells for 48 h. The best results were obtained for the hydrogel composed of 3% alginate, 7% gelatin, and 90% NaCl (0.9%).	[31]
Neuroblastoma cells	Alginate, carbon nanotubes, cellulose nanofibrils	Extrusion	Not reported	Electrical conductivity promoted the differentiation of SH-SY5Y human neuroblastoma cells into mature neural cells.	[27]
Osteosarcoma cells	Chitosan, gelatin	Extrusion	6 × 10^6^ cells/mL	The hydrogel supported growth the growth of UMR-106 rat osteosarcoma cells, but fragmentation of the constructs appeared after 14 days.	[37]
Pancreatic cancer cells, fibroblasts	PEG	Extrusion	1 × 10^7^ cells/mL	The scaffold’s modularity was demonstrated for pancreatic ductal adenocarcinoma and human dermal fibroblast cells.	[77]

**Table 6 gels-10-00008-t006:** Recent applications of 3D-printed hydrogels in drug screening and toxicity testing.

Disease	Hydrogel Composition	Technique	Cells	Tested Substances	In Vitro Results	Ref.
Angiogenesis	GelMA	Photopolymerization	HUVECs, 3 × 10^6^ cells/mL	Bevacizumab	A drug-screening chip composed of HUVECs that sprout in response to VEGF was established. Bevacizumab, an anti-VEGF antibody, was shown to inhibit the sprouting of HUVECs after 3 days of perfusion.	[166]
Brain cancer	Methacrylated collagen, thiolated hyaluronic acid	Photopolymerization	Patient-derived glioblastoma (GBM) cells, 8 × 10^6^ cells/mL	NSC59984, temozolomide	The printed organoids supported the growth of patient-derived GBM cells. A dose-dependent response to NSC59984, an experimental p53 activator compound, and temozolomide, the most frequently prescribed drug for patients with brain tumors, was demonstrated.	[46]
Brain cancer, prostate cancer	Alginate, gelatin	Extrusion	DU145 prostate cancer cells, U87 GBM cells, 5 × 10^4^ cells/mL	Dasatinib	The scaffold supported the growth of DU145 and U87 cells. The cells were significantly more resistant to dasatinib, a tyrosine kinase inhibitor, than corresponding monolayer-cultured cells.	[167]
Brain cancer, soft tissue cancer	Collagen methacrylate, hyaluronic acid	Photopolymerization	Patient-derived GMB and sarcoma cells	Dacomitinib, doxorubicin, imatinib, NSC59984	The scaffold supported the growth of patient-derived GBM and sarcoma cells. It also allowed the proof-of-concept screening of drugs against GBM (i.e., dacomitinib, NSC59984) and sarcoma cells (i.e., doxorubicin, imatinib).	[47]
Breast cancer	GelMA	Photopolymerization	MDA-MB-231 cells, 2–8 × 10^6^ cells/mL	Epirubicin, paclitaxel	The scaffold maintained the viability and ability of MDA-MB-231 cells to spread. The cells in the hydrogel had a higher viability against epirubicin and paclitaxel.	[168]
Breast cancer	Alginate, gelatin	Extrusion	MCF-7 cells, 3 × 10^6^ cells/mL	Camptothecin, paclitaxel	The scaffolds were used to compare the resistance of MCF-7 and a CD44-positive subset against camptothecin and paclitaxel.	[169]
Liver cancer	GelMA	Photopolymerization	HepG2/C3A cells, HUVECs, 2–4 × 10^6^ cells/mL	Sorafenib	The endothelialized liver lobule-like constructs were used for sorafenib screening, and stronger drug resistance was obtained when compared to hepatocyte spheroids alone.	[170]
Liver toxicity	Alginate, pluronic F127	Extrusion	HepG2/C3A cells, 2–2 × 10^6^ cells/mL	Acetaminophen	The cells demonstrated high viability and liver-specific metabolic activity in the scaffolds. Compared to 2D cultures, the cells in 3D constructs exhibited an increased sensitivity to a well-known hepatotoxic drug, acetaminophen.	[165]
Lung cancer	Vitrogel	Extrusion	HCC827 cells, MDA-MB-231 cells, and on-small cell lung cancer cells (NSCLC), 5 × 10^4^ cells/scaffold	Docetaxel, doxorubicin, erlotinib	The 3D scaffolds supported the rapid growth of spheroids, and IC50 values demonstrated higher resistance of 3D-cultured HCC827 lung adenocarcinoma cells, MDA-MB-231 cells, and patient-derived non-small cell lung cancer cells (NSCLC) to docetaxel, doxorubicin, and erlotinib compared to 2D monolayers.	[171]
Respiratory infection	Alginate, collagen, gelatin, alveolar epithelial cells	Extrusion	A549 cells, fibroblasts, and THP-1 cells, 3.5 × 10^6^–2.5 × 10^7^ cells/scaffold	Oseltamivir	The printed lung model was composed of A549 human lung adenocarcinoma cells, normal human fibroblasts, and macrophage-like THP-1 cells. The administration of neuraminidase inhibitor oseltamivir inhibited the growth of influenza A virus.	[48]

**Table 7 gels-10-00008-t007:** Recent applications of 3D-printed hydrogels for drug delivery.

Disease	Hydrogel Composition	Technique	Model	Physicochemical Properties	In Vitro Efficacy	In Vivo Efficacy	Ref.
Bone cancer	Calcium phosphate, chitosan, chondroitin sulfate, methacrylated hyaluronic acid, colony-stimulating factor 1 receptor inhibitor (GW2580)	Photopolymerization	Mouse	Pore size: ~500 μm Release: ~90% of GW2580 over 120 h	The scaffold inhibited the proliferation and osteoclastogenesis of bone-marrow derived monocytes and reduced M2 macrophage polarization.	The scaffold reduced the growth of subcutaneously implanted 4T1 tumors after 9 days. It also reduced the polarization of macrophages to an M2 phenotype.	[175]
Bone defects	Alginate, BMP2 plasmid	Extrusion	Rat	Pore size: 200–400 μm	The gene-activated scaffold increased the BMP-2 secretion and osteogenic differentiation of MSCs.	After 8 weeks, the scaffold significantly increased new bone volume formation in the cranial defects (BV/TV: ~46%).	[176]
Bone defects	Alginate, gelatin, nano-attapulgite, leonurine hydrochloride	Extrusion	Rat	Pore size: 10–500 μm Swelling ratio: ~5 Release: ~15% of leonurine hydrochloride over 800 h	The scaffold enhanced the osteogenic differentiation of MSCs and the vascularization of HUVECs.	The drug-loaded scaffold enhanced both angiogenesis and bone formation in rat skull defects (BV/TV: ~17%).	[177]
Bone defects	Hyaluronic acid methacrylate, bisphosphonates	Photopolymerization	Rat	Compressive modulus: 50 kPa Release: 10–13% release of bisphosphonates over 7 days	The hydrogel can release bisphosphonates in response to acidic pH. It inhibited the osteoclastic differentiation of macrophages and promoted the apoptosis of osteoclasts.	After 8 weeks, the bisphosphonate-loaded scaffold enhanced in situ bone regeneration in calvarial defects (BV/TV: ~14%).	[178]
Bone defects	GelMA, reduced glutathione	Photopolymerization	Mouse	Pore size: ~500 μm Swelling ratio: ~3 Compressive modulus: 3.5 kPa Release: ~400 μM of reduced glutathione over 15 days	The scaffold enhanced the proliferation, migration, and osteogenic differentiation of MC3T3-E1 cells.	After 8 weeks, the implantation of the scaffold into the calvarial defects of diabetic rodents resulted in enhanced bone regeneration (BV/TV: ~45%).	[179]
Bone defects	GelMA, PCL, osteoblasts, Wnt agonist (CHIR99021)	Photopolymerization	In vitro	Not evaluated	The scaffold promoted the osteogenic differentiation of ST2 cells and tube formation of HUVECs.	Not evaluated	[180]
Cartilage defects	dECM, GelMA, hyaluronic acid methacrylate, PCL, PLGA, platelet-derived growth factor-BB (PDGF-BB), kartogenin	Photopolymerization	Rabbit	Pore size: 80–1000 μm Compressive modulus: 12.99 MPa Release: ~70% and ~60% of PDGF-BB and kartogenin over 40 days, respectively	The scaffold enhanced the proliferation and chondrogenic differentiation of MSCs.	After 6 months, the drug-loaded scaffolds substantially boosted in vivo neo-meniscal regeneration in knee cartilage defects.	[181]
Pancreatic cancer	Methacrylated alginate, gemcitabine	Photopolymerization	Mouse	Pore size: ~3 mm Tensile modulus: 0.17 MPa Release: 20–40% of gemcitabine over 10–150 h.	The scaffolds inhibited the growth of MIA-PaCa-2 human pancreatic cancer cells.	After 4 weeks, the patches containing gemcitabine inhibited tumor growth without causing severe toxicity.	[182]
Retinal disease	PVA, fluorescein isothiocyanate conjugate-albumin (FITC-albumin)	Extrusion	Rabbit	Release: ~100% of FITC-albumin release over 200 days.	Not evaluated	In vivo implantation of the scaffold into the sclera of rabbits for 2 weeks revealed that FITC-albumin reached the target tissues of the retina and choroid.	[82]
Spinal cord injury	β-cyclodextrin, GelMA, neural stem cells, O-GlcNAc transferase inhibitor (OSMI-4)	Photopolymerization	Rat	Pore size: 100–150 μm Swelling ratio: 11 Release: 80% of OSMI-4 after 72 h	The scaffold supported the differentiation of neural stem cells into mature neurons.	After 8 weeks, the scaffold stimulated neuronal regeneration and axonal growth, resulting in a significant recovery of locomotor function in spinal cord injury models (Basso–Beattie–Bresnahan (BBB): 7.5).	[33]
Spinal cord injury	β-cyclodextrin, gelatin, spinal cord dECM, PCL, PEGDA, secreted frizzled-related protein 1 inhibitor (WAY-316606)	Photopolymerization	Rat	Pore size: 10–20 μm	The scaffold supported the differentiation of neural stem cells into mature neurons	After 8 weeks, the composite hydrogel considerably improved rat motor function after induction of spinal cord injury (BBB: 18.4).	[34]
Wounds	[2-(acryloyloxy) ethyl] Trimethylammonium chloride, GelMA, polyurethane, doxycycline	Extrusion	Rat	Pore size: 2 mm Compressive modulus: ~45 kPa Release: ~90% of doxycycline over 5 h	The scaffold promoted the migration of HUVECs and the M2 polarization of macrophages.	After 8 weeks, the drug-loaded scaffold promoted wound healing (WC: ~100%) by decreasing ROS and inflammation.	[183]
Wounds	GelMA, curcumin	Photopolymerization	Mouse	Pore size: ~200 μm Porosity: ~65% Swelling ratio: ~2 Young’s modulus: ~110 kPa	The scaffold reduced ROS generation and apoptosis in MSCs.	After 21 days, the scaffold containing curcumin promoted cell survival and accelerated in vivo diabetic wound healing (WC: ~95%).	[184]

**Table 8 gels-10-00008-t008:** Recent applications of 3D-printed hydrogels for the development of responsive materials.

Stimulus	Hydrogel Composition	Technique	Model	Physicochemical Properties	In Vitro and In Vivo Efficacy	Ref.
Calcium concentration	Alginate, nHA, silicon quantum dots	Extrusion	Rabbit	Pore size: 200–400 μm Swelling ratio: 1–1.2	The Ca^2+^-rich urine reactivated the crosslinking of the scaffolds. Higher Ca^2+^ concentration promoted stiffness elevation, which contributed to the repair of the urethra in vivo after 8 weeks.	[187]
Irradiation	Alginate, BSA, copper sulfide nanoparticles	Extrusion	Mouse	Pore size: ~100 μm	Under 808 nm irradiation, the scaffolds demonstrated efficient photothermal, photodynamic, and chemodynamic effects against 4T1 tumors in vitro and in vivo.	[111]
Irradiation	Alginate, gelatin, copper oxide nanoparticles	Extrusion	Mouse	Pore size: ~50 μm Swelling ratio: 17.4 Release: ~60 μg of Cu^2+^ over 24 h	The copper oxide nanoparticles released ions that generated intracellular ROS and acted as photothermal agents. After 10 days, the recurrence of H22 murine hepatocellular carcinoma after primary resection was suppressed.	[110]
Irradiation	Alginate, MXenes, PNIPAM, VEGF	Photopolymerization	Mouse	Pore size: ~500 μm Release: ~600 pg of VEGF over 5 days	The scaffolds demonstrated an NIR-responsive shrinkage/swelling behavior, which facilitated the controlled release of VEGF. After 9 days, the scaffolds enhanced skin flap survival by promoting angiogenesis, reducing inflammation, and inhibiting apoptosis.	[95]
Irradiation	Gelatin, β-TCP, wesselsite, doxorubicin	Extrusion	Rat	Pore size: 0.5–1 mm Porosity: 60–70% Release: 60% of doxorubicin over 80 min	The scaffold generated hyperthermia that could induce the gel-sol transition of the gelatin, which triggered on-demand doxorubicin release. The scaffold inhibited the proliferation of MG-63 osteosarcoma cells.	[185]
Irradiation	Gelatin, silk fibroin, methylene blue-loaded UiO-66 nanoparticles	Extrusion	Mice	Pore size: 20–500 μm Compressive strength: 20–125 kPa	The scaffold demonstrated antibacterial properties from the photodynamic therapy effect of the nanoparticles and promoted the healing of infected wounds after 14 days (WC: ~95%).	[188]
Irradiation	GelMA, zinc oxide microparticles, VEGF	Photopolymerization	Mouse	Pore size: ~100 μm Compressive strength: 80–150 kPa	The patches, which encapsulated VEGF and antibacterial tetrapodal zinc oxide, can be activated through ultraviolet/visible light exposure. The scaffolds showed less immunogenicity and enhanced wound healing in mice after 14 days (WC: ~95%).	[114]
Magnetic field	Chitosan, citric acid-coated superparamagnetic iron oxide nanoparticles	Extrusion	In vitro	Swelling ratio: 2.29	The untethered milli-gripper could successfully grasp a cargo, transfer it to the desired position, and then release it under the influence of an applied magnetic field.	[113]
Magnetic field	GelMA, neodymium-iron-boron magnetic beads	Photopolymerization	In vitro	Swelling ratio: 1.6 Young’s modulus: ~100 kPa	A magnetically deformable scaffold was developed so that cells can be examined in an environment that replicates the dynamic, curved, and flexible characteristics of living tissues.	[112]
Magnetic field	GelMA, PEGDA, iron oxide nanoparticles	Photopolymerization	Mouse	Release: 1.4 mM of acetylsalycilic acid over 60 h, 70 μg/mL of doxorubicin over 30 h	The scaffold can respond to magnetic fields for actuation and drug delivery. In vivo studies indicate that the microrobots carrying acetylsalycilic acid and doxorubicin inhibited the growth of HeLa cells after 14 days.	[78]
Temperature	PNIPAM	Photopolymerization	In vitro	Pore size: ~2 mm	The thermosensitivity of the hydrogels endowed the scaffold with reversible enhancement of resolution, while the supramolecular crosslinking provided the benefit of on-demand disintegration.	[93]
Temperature	Hyaluronic acid, PNIPAM, gentamicin, vancomycin	Photopolymerization	Sheep	Release: 100% of gentamicin after 336 h, 100% of vancomycin after 336 h	The thermoresponsive scaffold laden with gentamicin and vancomycin outperformed current clinical practice in eradicating chronic methicillin-resistant *Staphylococcus aureus* orthopedic infection in sheep.	[189]
Ultrasound	GelMA, perfluorohexane nanodroplets with hemoglobin	Photopolymerization	Rat	Pore size: ~20 μm Release: ~6 mg/L of dissolved oxygen at 200 s	Ultrasound-controlled oxygen release within the nanodroplets improved the oxygen supply in the cardiac patch, which increased cell viability within the patch and enhanced the left ventricular ejection fraction in mice after 14 days.	[186]
Ultrasound	GelMA, PEGDA, gold-nanoparticle-decorated tetragonal barium titanate, VEGF	Photopolymerization	Rat	Pore size: ~1 mm Swelling ratio: 1.5–2	Under the influence of ultrasound, the piezoelectric hydrogel patch exerted significant infection elimination activity. Its sustained release of growth factors enhanced wound healing after 10 days (WC: ~95%).	[79]

## Data Availability

Not applicable.

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
