# Peer review of "Current Biomedical Applications of 3D-Printed Hydrogels"

_gels, 2023, doi:10.3390/gels10010008_

Round 1
Reviewer 1 Report
Comments and Suggestions for Authors
The manuscript is well organized and well written. There are only several minor errors or inconsistencies that are specified below.
- Table 1: DLP stands for "Digital Light Processing" (i.e., not "Direct Light Processing")
- line 262: Correct chemical name of GelMa would be either: "methacryloylgelatin" or: "gelatin methacrylate".
- lines 322/323: less ambiguous name of PLA would be: poly(lactic acid)
- line 330/331: there is: lactic acid monomers -> there should be: lactide, which is a cyclic lactic acid dimer
- lines 332,336: polymerization -> copolymerization
- line 364: of the nitrogen -> on the nitrogen
- line 365: delete: cationic, (because PEDOT is not a cationic polymer)
- lines 445,770,779 and Table 8(4x): electromagnetic field -> magnetic field (because "electromagnetic fields" are variable fields, usually alternating at high frequencies. In order to move a paramagnetic particle from one location to another, a static magnetic field has to be applied, that has no electrical field component, so it is not an electromagnetic field)
- line 578: there is: quaternized chitosan-g-polyaniline chitosan. The second "chitosan" is probably redundant.
- Table 3, ref.[54]: The hydrogel composition is incomplete, because dECM alone does not photopolymerize. Same applies to Table 5, ref.[22].
- Table 5: The acronym RGD should be defined somewhere in the text.
- Table 7: For brevity, the name "poly(lactic-co-glycolic) acid" in Table 7 may be replaced with PLGA acronym that was defined previously in line 323.
- Table 8: The hydrogel composition taken from ref.[181] does not contain a photopolymerizable component, so the "Photopolymerization" technique does not match the composition. Either the composition is incomplete or the "Photopolymerization" trchnique is inappropriate.
- line 859: the facilitate -> to facilitate
Hence, in my opinion, the manuscript is acceptable for publication as is, assuming that the authors will correct the above mentioned minor errors on the article proof.
Author Response
We thank the reviewer for providing an elaborate review on our manuscript. The edits/suggestions requested were very constructive and have shaped our manuscript in a better way. We have made tremendous efforts to address each and every suggestions and have edited our manuscript in accordance with the comments. Please refer to attached document containing our responses to the individual comments. We have taken extreme care to incorporate all requested changes in to the manuscript. Thanks again for providing us such constructive comments.

Reviewer 2 Report
Comments and Suggestions for Authors
I have nothing to add.
Honestly, it's easier for everyone to understand if there's an illustration.
Author Response
Thank you for taking time to review our manuscript and provide your feedback. Since this is a review article, we wanted it to be thorough with regards to current research in this arena and so we have only provided one illustration that totally depicts various fields of the 3D printing where gels are predominantly used. We have incorporated elaborate tables for every subsection (application) within this field. We thank you for providing us this comment and hope you could suggest accepting the manuscript in its current state.

Reviewer 3 Report
Comments and Suggestions for Authors
This review is devoted to 3D printing of hydrogels for bioapplications. Paper is well structured and possess scientifical soudness. Especially I would like to note authors' description of 3D printing part which is well organized and presented. Please, change table 1 to avoid full copypast from ref 7.
However, hydrogel part require more attention for readable presentation.
- on line 256 GelMa is mentioned. GelMa is seminatural-derived polymer, while subsecion is called 2.3.1 Natural polymers. Authors should write about different type of classification of polymers (not only plant-derived and animal derived) and suggest more clear presentation of 2.3.1 to avoid inaccuracies. Also it's better use "nature-derived" instead "natural"
- line 195, "These polymers include cellulose, starch, and cyclodextrins." First, ref is required, second, hemicelluloses are missed.
- lines 295-299. The sentence is too general, be more specific and give more concrete examples. There are alot of synthetic polymers used for bioapplications, i.e. PEG, polypropylene, PVP, block-copolymers
- line 349. PNIPAM demonstrates LSCT in water solutions, add this clarification. PNIPAM has'nt LSCT in DMF or 100% EtOH
- for section 2.31. and 2.3.2 FDA and other regulation rules should be mentioned. It's important for bioapplications and give understanding how fast materials could be applied in practise
- section 2.31 and 2.32 should have thier oun subsection for easyness of presentation
- section about mechism of gel formation (even short section) should be given). At least describe physical and chemical gelation 10.1039/D3CS00387F, 10.1002/adfm.202002759
- section 3.1 refs should be given. 3D bioprinting is very large field, there are a lot of reviews about tissue printing
- Tabled 3-7 should be rewritten. It seems that in column "result" authors just pasted conslusions from respective references. You should choose specific parameter (cell viability, mechanical or transport properties or other) to compare hydrogels based on choosed parameter.
- Please, update introduction with refs avout hydrogels and 3D printing https://doi.org/10.1016/j.jcis.2022.12.106, 10.1016/j.compositesb.2022.109863, 10.1016/j.actbio.2019.05.032, 10.1002/adfm.202009432, 10.1088/1748-605X/ac7308
Comments on the Quality of English LanguageMinor editing of English language required
Author Response
We thank the reviewer for providing an elaborate review on our manuscript. The edits/suggestions requested were very constructive and have shaped our manuscript in a better way. We have made tremendous efforts to address each suggestion and have edited our manuscript in accordance with the comments. Please refer to points below containing our responses to the individual comments. We have taken extreme care to incorporate all requested changes into the manuscript. Thanks again for providing us such constructive comments

Round 2
Reviewer 3 Report
Comments and Suggestions for Authors
Authors addressed some questions issued by reviewers. However, tables 2-7 are poorly written. Authors need to come up with a parameter and find its data in each work and not just copy 1-2 sentence from conclusions of each cited article. In this form it's not a "review" style its more "table of content" style, which is not scientific. After correcting the tables 2-7, the manuscript can be published.
Comments on the Quality of English LanguageMinor editing is required
Author Response
Dear Reviewer, thank you so much for taking time to review our manuscript and providing us such constructive comment. We now have totally modified the table 2-7 and now included more specific details including physicochemical properties, in vitro efficacy and in vivo efficacy. We tried to make this as elaborate as possible and provided the gist of each column as briefly and effectively as possible.
We have attached revised manuscript with updated table.
We now believe that this manuscript is thorough and hope you recommend publishing in its current state.

Round 3
Reviewer 3 Report
Comments and Suggestions for Authors
Authors addressed all listed by reviewers issues. I recommend to accept this review for publication
Comments on the Quality of English LanguageMinor English editing is required